# Modified Sulfanilamide Release from Intelligent Poly(*N*-isopropylacrylamide) Hydrogels

**DOI:** 10.3390/pharmaceutics15061749

**Published:** 2023-06-16

**Authors:** Ana Dinić, Vesna Nikolić, Ljubiša Nikolić, Snežana Ilić-Stojanović, Stevo Najman, Maja Urošević, Ivana Gajić

**Affiliations:** 1Faculty of Technology, University of Niš, Bulevar Oslobođenja 124, 16000 Leskovac, Serbia; anatacic@tf.ni.ac.rs (A.D.); nikolicvesna@tf.ni.ac.rs (V.N.); snezanai@tf.ni.ac.rs (S.I.-S.); maja@tf.ni.ac.rs (M.U.); ivana@tf.ni.ac.rs (I.G.); 2Department of Biology and Human Genetics, Faculty of Medicine, University of Niš, Blvd. Dr Zorana Djindjica 81, 18108 Niš, Serbia; stevo.najman@medfak.ni.ac.rs; 3Department for Cell and Tissue Engineering, Faculty of Medicine, University of Niš, Blvd. Dr Zorana Djindjica 81, 18108 Niš, Serbia

**Keywords:** sulfanilamide, *N*-isopropylacrylamide, thermosensitive hydrogels, swelling kinetics, characterization, release kinetics

## Abstract

The aim of this study was to examine homopolymeric poly(*N*-isopropylacrylamide), p(NIPAM), hydrogels cross-linked with ethylene glycol dimethacrylate as carriers for sulfanilamide. Using FTIR, XRD and SEM methods, structural characterization of synthesized hydrogels before and after sulfanilamide incorporation was performed. The residual reactants content was analyzed using the HPLC method. The swelling behavior of p(NIPAM) hydrogels of different crosslinking degrees was monitored in relation to the temperature and pH values of the surrounding medium. The effect of temperature, pH, and crosslinker content on the sulfanilamide release from hydrogels was also examined. The results of the FTIR, XRD, and SEM analysis showed that sulfanilamide is incorporated into the p(NIPAM) hydrogels. The swelling of p(NIPAM) hydrogels depended on the temperature and crosslinker content while pH had no significant effect. The sulfanilamide loading efficiency increased with increasing hydrogel crosslinking degree, ranging from 87.36% to 95.29%. The sulfanilamide release from hydrogels was consistent with the swelling results—the increase of crosslinker content reduced the amount of released sulfanilamide. After 24 h, 73.3–93.5% of incorporated sulfanilamide was released from the hydrogels. Considering the thermosensitivity of hydrogels, volume phase transition temperature close to the physiological temperature, and the satisfactory results achieved for sulfanilamide incorporation and release, it can be concluded that p(NIPAM) based hydrogels are promising carriers for sulfanilamide.

## 1. Introduction

Hydrogels are weakly cross-linked, hydrophilic polymeric materials that have the ability to bind and retain large amounts of water or biological fluids [1], because they contain hydrophilic groups (hydroxyl, carboxyl, amide) in their structure [2]. They represent homopolymers or copolymers insoluble in water due to chemical or physical crosslinking, which provides three-dimensional, mesh structure and physical integrity. Hydrogels resemble living tissues more than any other class of synthetic biomaterials, primarily because of their high water content and soft consistency, which contributes to their biocompatibility [1,3]. In addition, hydrogels can be biodegradable, easily synthesized, have a high incorporation efficiency of various active substances and cells, and have low toxicity [4,5], which makes them suitable for application in drug delivery.

Hydrogels can be classified based on parameters such as origin, number of monomers, charge of the side groups, physical properties, and crosslinking mode [1,6]. The classification of hydrogels according to their reaction to a change in environmental conditions is particularly significant, with conventional hydrogels that do not respond to external stimuli and hydrogels sensitive to external stimuli, i.e., “intelligent” hydrogels. Stimuli-sensitive hydrogels have the ability to respond to small changes in environmental conditions by drastically changing swelling properties, network structure, permeability, or mechanical properties [1,7]. Different hydrogels can respond to different stimuli, such as temperature, solvent polarity, pH, charge, light, electric field, the presence of certain biomolecules, such as enzymes and receptors, or their combination [8,9].

Temperature-sensitive polymers are perhaps the most examined class of stimuli-sensitive polymers for use in drug delivery. The use of temperature as a signal is justified by the fact that body temperature changes in the presence of various pathogens and pyrogens, and in conditions such as infections and inflammation. Therefore, a change in temperature can be a useful stimulus to begin releasing an antimicrobial, anti-inflammatory or antipyretic drug from various temperature-sensitive delivery systems. One of the most relevant thermosensitive polymers for use in biomedicine is poly(*N*-isopropylacrylamide), p(NIPAM), because it is biocompatible with tissues, stimulates cell growth, has satisfactory mechanical properties and the ability to easily interact with other polymers [10].

The linear p(NIPAM) has a lower critical solution temperature (LCST) at 32 °C [11]. The sharp phase transition of linear p(NIPAM) occurs due to conformational changes and is a consequence of the release of structured water molecules around the polymer chain [12]. Below the LCST, the polymer is in a hydrated state and highly soluble, while above the LCST it becomes more hydrophobic due to the disruption of hydrogen bonds, which leads to a change in solubility and to polymer aggregation [13]. The cross-linked p(NIPAM) hydrogels have a volume phase transition temperature (VPTT). At temperatures below the VPTT, the formation of hydrogen bonds between amide groups and water molecules is favored, due to which the hydrogel binds large amounts of water and swells. With temperature increase, hydrogen bonds weaken, and interactions between hydrophobic groups strengthen, which leads to the release of water molecules from the hydrogel structure. At the same time, there is a sudden collapse of polymer chains and a volume phase transition [13,14]. Thus, the LCST refers to the linear polymers, while the VPTT is a characteristic of cross-linked polymeric hydrogels, which swell in the presence of water, but do not dissolve, that is, they are always two-phase systems [15].

p(NIPAM) hydrogels belong to the negative thermosensitive hydrogels which contract upon heating, decreasing their volume [15]. At the physiological body temperature that is higher than the volume phase transition temperature of p(NIPAM) hydrogels, the intermolecular interactions between sulfanilamide and side groups of the polymer matrix become broken and contraction of the polymer matrix starts, which initiates drug release. Drug release from hydrogels can be diffusion controlled, swelling controlled or chemically controlled [16]. The rate of release and the amount of drug released from the hydrogel depend on the degree of swelling and crosslinking density of the hydrogel, as well as on the properties of the drug itself, such as molecular weight and charge [17]. In p(NIPAM)-based hydrogels, the initial rapid release of the active substance is followed by a much slower, controlled release [10]. That is another reason for using p(NIPAM)-based hydrogels as active substance carriers [18]. In direct and indirect contact with mouse fibroblasts, p(NIPAM) hydrogel does not reduce the viability of fibroblast cells and thus shows minimal cytotoxicity [19].

Sulfanilamide (4-aminobenzenesulfonamide) is the basic structural and functional unit of the entire class of antimicrobial sulfonamides. The chemical structure of sulfanilamide is given in Figure 1. Sulfonamides are the first synthesized, selectively toxic, broad-spectrum antimicrobials [20,21], whose application in therapy is partly limited by inadequate physicochemical properties, such as poor aqueous solubility and photosensitivity [22]. To overcome these disadvantages and enable the more effective and safer application of sulfonamide drugs, various advanced pharmaceutical systems can be used, which also enable targeted delivery and modified release of active substances. Sun et al. (2020) developed the alginate hydrogel fibers with sulfanilamide cross-linked using a combination of Ca^2+^ ions and glutaraldehyde for application as a wound dressing. It has been shown that sulfanilamide is released slowly and it achieves a wide spectrum of antibacterial activity, which is extremely important in wound healing [23]. Sulfanilamide-loaded alginate hydrogel fibers based on ionic and chemical crosslinking showed better properties than alginate hydrogel fibers cross-linked only by calcium ions [24]. El-Din and Ibraheim (2021) examined the effect of nanocomposite hydrogels prepared by gamma-radiation copolymerization of acrylic acid onto plasticized starch/montmorillonite clay/chitosan blends with sulfanilamide on wound healing [25]. Different copolymeric p(NIPAM) hydrogels, including copolymers with gelatin [17], glycidyl methacrylate (GMA), and *N*,*N*-dimethyl acrylamide [26] and acrylic acid [27] were tested as carriers for sulfanilamide.

Copolymer hydrogels have mainly been used for the delivery of some drugs: a hydrogel based on NIPAM and dextran was used as a carrier for the controlled release of ornidazole and ciprofloxacin [28], while nanoparticles made from hydrogels based on NIPAM, acrylic acid, and hydroxyethyl methacrylate formulated to achieve controlled release of amoxicillin [29]. Copolymer hydrogels of p(NIPAM) with gelatin [30], methacrylic acid [31] or allyl-amine [32] can be used as carriers for silver nanoparticles, which are used in antimicrobial therapy of wounds and burns. Copolymer hydrogels of NIPAM with 2-hydroxypropyl methacrylate were investigated as carriers for the controlled release of ibuprofen [33].

Lee and Lee synthesized hybrid hydrogels based on NIPAM and gelatin for encapsulation and release of sulfanilamide from hybrid hydrogels [17]. Lee and Huang investigated the release of sulfanilamide, as a model drug, from a copolymer hydrogel based on NIPAM, glycidyl methacrylate (GMA) and *N*,*N*-dimethylacrylamide (DMA) [26]. Lee and Cheng investigated the effect of monomer content on sulfanilamide release and properties of a biodegradable hydrogel based on NIPAM, acrylic acid, and poly(caprolactone)-diacrylate [27].

Thermosensitive bioadhesive hydrogels based on poly(*N*-isopropylacrilamide) and poly(methyl vinyl ether-*alt*-maleic anhydride) was used for the controlled release of metronidazole in the vaginal environment [34].

These researches were the motive to carry out tests of the homopolymer p(NIPAM) with sulfanilamide and define the possibility of making such formulations as well as their potential application.

The objective of this work was to examine, for the first time, homopolymeric p(NIPAM) hydrogels cross-linked with ethylene glycol dimethacrylate (EGDM) as carriers for sulfanilamide, as a model drug.

## 2. Materials and Methods

### 2.1. Reagents

For sulfanilamide and hydrogels synthesis, chlorosulfonic acid, 99% (Sigma Aldrich, Taufkirhen, Germany), acetanilide, 97% (Sigma Aldrich, Taufkirhen, Germany), hydrochloric acid (Alkaloid, Skoplje, Macedonia), ammonium hydroxide (Alkaloid, Skoplje, Macedonia), sodium hydroxide (Alkaloid, Skoplje, Macedonia), *N*-isopropylacrylamide, NIPAM, 99% (Acros Organics, Morris Plains, NJ, USA), 2,2′-azobis(2-methylpropionitrile), AIBN, 98% (Acros Organics, Morris Plains, NJ, USA) and ethylene glycol dimethacrylate, EGDM, 97% (Fluka Chemie AG, Buchs, Switzerland), were used. Methanol, 99.9% (Sigma-Aldrich GmbH, Steinheim, Germany), potassium bromide (KBr; spectroscopic purity) (Merck, Darmstadt, Germany), hydrochloric acid (HCl; ≥36.5%), sodium hydroxide (NaOH) and acetone, 99.5%, (Centrohem, Belgrade, Serbia) were used as well.

### 2.2. Sulfanilamide Synthesis and Purification

Sulfanilamide was synthesized from acetanilide and chlorosulfonic acid by using the conventional method, according to the previously described procedure [35]. Purification of the synthesized sulfanilamide was performed by recrystallization from aqueous solution. The purified sulfanilamide had a melting point of 163 °C, which is in accordance with the literature data [36].

#### ^1^H-NMR Spectrum of Sulfanilamide

^1^H-NMR spectrum of synthetized sulfanilamide was recorded in glass cuvette (5 mm diameter) on a Bruker AC 250E NMR spectrometer (Bruker Corporation, Billerica, MA, USA) at room temperature by using the method with multiple pulse repetitions. The operating frequency was 250 MHz. The sample was prepared by dissolving 500 µg of sulfanilamide in 200 µL of deuterated water, D_2_O.

### 2.3. Hydrogels Synthesis and Lyophilization

Hydrogels were synthesized according to the procedure described before [37]. In short, homopolymeric hydrogels were synthesized by free radical polymerization of NIPAM monomer using various content of the crosslinker EGDM (1, 1.5, 2 and 2.5 mol%). The amount of crosslinker was calculated in relation to the amount of monomer. The reaction of polymerization was initiated by adding 2.7 mol% of AIBN. Reaction mixtures prepared by dissolution of reagents in acetone and homogenization, were injected into the glass tubes, 5 mm in diameter, which were then sealed. Polymerization was thermally initiated in the following way: 0.5 h at 75 °C, 2 h at 80 °C and 0.5 h at 85 °C. This temperature profile of the hydrogel synthesis enables a slower formation of free radicals at the beginning of the reaction and a slower synthesis. Later, the temperature is increased to decompose the remaining amount of initiator and to increase the degree of monomer conversion. If the highest temperature is applied at the beginning of the polymerization process, larger amounts of radicals can be formed, which increases the possibility of side reactions. In order to remove the unreacted amounts of reagents, synthesized hydrogels were immersed in 30 cm^3^ of methanol for 72 h. After rinsing, hydrogels were dried to a constant mass in a drying oven at 40 °C.

Lyophilization of p(NIPAM) hydrogels in a swollen state was performed on the device Freeze Dryers Rotational-Vacuum-Concentrator (GAMMA 1-16 LSC, Osterode, Germany). Firstly, the swollen hydrogels were frozen at −40 °C for 24 h. Then, the solution amount was decreased by sublimation at −30 °C and pressure of 0.05 kPa for 12 h. Lastly, the hydrogels were heated at 20 °C under the pressure of 0.05 kPa for 6 h in order to remove the residual vapors. Lyophilized hydrogels were packed under a vacuum and stored in a refrigerator at 5 °C.

### 2.4. Residual Reactants Analysis

Methanolic solutions obtained by immersion of synthesized hydrogels were used for HPLC analysis in order to determine the residual monomer and crosslinker content. The analysis was performed on an Agilent 1100 Series HPLC device with a DAD 1200 Series detector (Waldborn, Fulda, Germany). A Zorbax Eclipse XDB-C18 column (250 × 4.6 mm, 5 µm) (Agilent Technologies, Inc., Santa Clara, CA, USA) was used and the detection wavelength was 225 nm. Methanol was used as an eluent with a flow rate of 1 cm^3^/min. The volume of injected samples was 10 µL, and the column temperature was 25 °C. The calibration curve for NIPAM was linear in the concentration range of 0.005–0.25 mg/cm^3^. Equation (1) applies with the linear correlation coefficient R^2^ = 0.9989:(1)A=25231 c+67.074

The calibration curve for EGDM was linear in the concentration range of 0.005–0.5 mg/cm^3^. The Equation (2) applies with the linear correlation coefficient R^2^ = 0.9987:(2)A=6864.4 c+20.97

In the Equations (1) and (2), *A* is the peak area (mAU·s) and *c* is the content of NIPAM and EGDM (mg/cm^3^), respectively.

### 2.5. Swelling of Hydrogels

The swelling of hydrogels was monitored gravimetrically by immersing the dry hydrogel samples (xerogels) in aqueous solutions of a certain pH (a solution of hydrochloric acid pH 2.2 or sodium hydroxide pH 7.4) and temperature, and then measuring their masses in certain time intervals, until equilibrium was reached. The solution excess was removed from the hydrogel surfaces using filter paper. Based on the samples masses before and after swelling, the swelling degree of hydrogels (α) was determined according to Equation (3):(3)$$\alpha =\frac{m-m_{0}}{m_{0}}$$
where *m*_0_ is the mass of the dry samples, and *m* is the mass of swollen hydrogels in the moment of time *t*.

To analyze the type of solvent diffusion process inside hydrogels, Equation (4) applies, which is valid for the initial phase of swelling (*M_t_*/*M_e_* ≤ 0.6) [38]:(4)$$\frac{M_{t}}{M_{e}}=k\;\cdot \;t^{n}$$
where *M_t_* is the mass of the absorbed solvent at time *t*, *M_e_* is the mass of the absorbed solvent in equilibrium, *k* is the constant that is characteristic for a certain type of polymer network (min^1/*n*^) and *n* is the diffusion exponent.

To determine diffusion coefficient, *D*, Equation (5) applies: (5)$$\frac{M_{t}}{M_{e}}=4{\left(\frac{Dt}{\pi l^{2}}\right)}^{1/2}$$
where *D* is a diffusion coefficient (cm^3^/min) and *l* is the thickness of the dry hydrogel (cm). By converting the exponential Equation (5) to a logarithmic form, Equation (6) is obtained (a linear relationship between ln(*M_t_/M_e_*) and ln*t*, and the diffusion coefficient *D* is calculated from the graph intercept:(6)$$\ln\left(\frac{M_{t}}{M_{e}}\right)=\frac{{4D}^{1/2}}{\pi^{1/2}l}+\frac{1}{2}(\ln t)$$

### 2.6. The Examination of p(NIPAM) Hydrogels as Carriers for Sulfanilamide

#### 2.6.1. Sulfanilamide Incorporation into the Hydrogels

In order to incorporate sulfanilamide into the polymeric network, synthesized xerogels (~0.05 g) were swelled in the sulfanilamide methanolic solution concentration 50 mg/cm^3^ for 48 h at room temperature. The available amount of sulfanilamide for incorporation into the hydrogel was 500 mg/g_xerogel_. After reaching equilibrium, the swollen p(NIPAM) hydrogels with incorporated sulfanilamide were separated from the solution by decanting. The hydrogel samples were washed using distilled water to remove excess sulfanilamide. The masses of samples before and after swelling in the sulfanilamide solution were measured in order to calculate the loading efficiency. Loading efficiency (*η*) of sulfanilamide was calculated using Equation (7):(7)$$\eta (\%)=\frac{L_{g}}{L_{u}}\;\cdot \;100$$
where *L_g_* is the mass of sulfanilamide incorporated into the hydrogel (mg/g_xerogel_) and *L_u_* is the maximum sulfanilamide mass available for incorporation (mg/g_xerogel_). Hydrogel with sulfanilamide was dried from methanol and then put in contact with the amount of water that it can absorb, according to the swelling degree, in order to avoid premature release of sulfanilamide. After that, the release of sulfanilamide from the hydrogels in swollen state was monitored.

#### 2.6.2. In Vitro Sulfanilamide Release from Hydrogels

Sulfanilamide release from hydrogels was studied under the simulated physiological conditions. The swollen hydrogels with incorporated sulfanilamide were soaked in 3 cm^3^ of adequate medium (a solution of hydrochloric acid pH 2.2 or sodium hydroxide pH 7.4) and tempered in a water bath at 20 ± 1 or 37 ± 1 °C for 24 h. Aliquots of the medium with released sulfanilamide were taken in certain time intervals, diluted with methanol and analyzed using HPLC method. The analysis was performed on the Agilent 1100 Series device under the following conditions: detector DAD 1200 Series; detection wavelength 258 nm; column Zorbax Eclipse XDB-C18 (250 × 4.6 mm, 5 µm); eluent methanol; eluent flow 1 cm^3^/min; sample injection volume 10 µL; column temperature 25 °C. Under the defined analysis condition, the calibration curve for sulfanilamide was linear in the concentration range of 0.005–0.05 mg/cm^3^. The Equation (8) applies with the linear correlation coefficient R^2^ = 0.9999: (8)A=119615.7 c+76.27
where *A* is the peak area (mAU·s) and *c* is the sulfanilamide concentration (mg/cm^3^).

#### 2.6.3. Fourier Transform Infrared Spectroscopy (FTIR)

FTIR spectra of sulfanilamide, synthesized xerogels and xerogels with incorporated sulfanilamide were recorded using thin transparent pastilles made of 1 mg of sample and 150 mg of potassium bromide (KBr, 99%, Merck, Darmstadt, Germany) by vacuuming and pressing under the pressure of 200 MPa. The FTIR spectra of all samples were recorded on the BOMEM MB-100 spectrophotometer (Hartmann & Braun, Quebec, QC, Canada) in the area of wavenumbers from 4000 to 400 cm^−1^ and analyzed using Win-Bomem Easy software 1.9.11.

#### 2.6.4. X-ray Diffraction (XRD)

Grounded samples of xerogels and xerogels with incorporated sulfanilamide were analyzed on diffractometer for powder Philips PW1030 using monochromatic CuK_α_ radiation in the range of 2*θ* = 5–70° with 0.05° step and recording time of τ = 5 s. The operating device voltage and electric current were 40 kV and 20 mA, respectively.

#### 2.6.5. Scanning Electron Microscopy (SEM)

The morphology of lyophilized hydrogels and lyophilized hydrogels with incorporated sulfanilamide was analyzed using scanning electron microscopy. Grounded samples were coated with a gold/palladium (85/15) alloy under vacuum using JEOL Fine Coat JFC 1100E Ion Sputter (JEOL Co., Tokyo, Japan) and scanned on a JEOL Scanning Electron Microscope JSM-5300 (JEOL Co., Tokyo, Japan).

### 2.7. Statistical Analysis of Data

Analysis of variance (ANOVA) was used to determine the significance (*p* ≤ 0.05) of the data obtained in the experiments. All results were determined to be within the 95% confidence level for reproducibility.

## 3. Results and Discussion

### 3.1. The Residual Reactants Analysis

In order to determine the residual reactants content in the synthesized hydrogels, HPLC analysis of methanolic solutions obtained by hydrogel immersion was performed. Under the applied analysis conditions, the chromatographic peaks of the monomer and crosslinker were well separated, with retention times at R_t_ = 2.478 min and R_t_ = 2.700 min, respectively (Figure 2).

The content of unreacted reactants in the samples of synthesized homopolymeric hydrogels (mg/g hydrogel), as well as their share in relation to the initial mass in the reaction mixture, are shown in Table 1.

The residual monomer and crosslinker molecules can alter the mechanical properties of the material, and they are often harmful components, which can cause skin, eye, and respiratory tract irritation, as well as allergic reactions including contact dermatitis [39], so their content in polymers must be carefully controlled to keep the polymer safe for use. The content of residual NIPAM monomer ranged from 0.95–1.91%, while the range is narrower for the crosslinker content (1.29–1.88%), relative to their initial content in the reaction mixture. The presented data showed that most (>98%) of monomers and crosslinker were used in polymer synthesis, and that the residual amount of reactants was within tolerance limits [40].

However, in order for the synthesized hydrogels to be safely used as carriers of active substances, the residual amounts of reactants have to be removed from the hydrogels. For this purpose, hydrogels were treated with methanol (0.5 g of hydrogel was soaked in 30 cm^3^ of methanol), according to the procedure developed earlier [37]. After methanol treatment, the hydrogels were immersed in the methanol/distilled water solutions 75%/25%, 50%/50%, 25%/75%, and 0%/100%, *v*/*v*, during 24 h to gradually rinse the methanol from the synthesized hydrogels. The hydrogels were dried for about 3 h at 40 °C to the constant mass. After the extraction and drying, it was not possible to detect residual amounts of reactants in the synthesized hydrogels.

### 3.2. Swelling Study

The hydrogel swelling capacity depends on several factors, such as the chemical structure of monomers and the crosslinking degree of the polymer network, the concentration, charge, and pKa values of ionizable groups [6], as well as the characteristics of the swelling medium including temperature and pH value. In order to determine how some of these parameters (pH value, temperature, and the crosslinking degree) influence the swelling properties of synthesized hydrogels, swelling of p(NIPAM) hydrogels with different crosslinker content was followed in the solutions of pH 2.2 and 7.4 at the temperature of 20 ± 1 and 37 ± 1 °C.

The dependence of the swelling degree on time, for p(NIPAM) hydrogels with different crosslinker content at 20 ± 1 °C and pH 2.2 and 7.4, respectively, is shown in Figure 3.

In all tested samples, the swelling degree values increased sharply during the first 300 min and then gradually, until equilibrium was reached. The hydrogel with 1.5 mol% of crosslinker had the highest swelling degree (α = 30.55) at pH = 7.4. The same hydrogel had a swelling degree of α = 29.14 at pH = 2.2. The swelling degree values obtained for certain p(NIPAM) hydrogels at different pH were similar, indicating that homopolymeric p(NIPAM) hydrogels are not sensitive to changes in pH, which is in agreement with their chemical structure and the results of other authors [41]. The p(NIPAM) hydrogels belong to nonionic hydrogels, as they do not contain functional groups susceptible to ionization, so they do not react to changes in the pH value of the environment. The p(NIPAM) hydrogel with 1 mol% of crosslinker had lower swelling degrees than hydrogels with 1.5 mol% of crosslinker as the sufficient crosslinking density of hydrogels was not reached due to the low crosslinker concentration. This apparent anomaly can be explained by the fact that in the case of hydrogels with 1% of crosslinker, a polymer network with too long branches between the nodes was obtained. In this way, some parts of the polymer network did not contribute significantly to the swelling. On the other hand, the hydrogel with the highest crosslinker content (2.5 mol%) had the lowest swelling degree at both pH values, α = 17.58 (pH = 7.4) and α = 15.71 (pH = 2.2). Based on the presented results, it can be concluded that the increase in crosslinker content reduced the swelling degree of hydrogels [42,43], which confirms that the hydrogel composition affects its swelling characteristics. As the content of the crosslinker increases, the crosslinking density of the hydrogels also increases, while the mobility of polymer chains and the branches length decreases, so hydrogels with higher crosslinker content have a lower ability to absorb water, leading to a lower swelling degree.

In addition to the crosslinker content, the crosslinker type and the polymerization method also affect the swelling degree of p(NIPAM) hydrogels [42,43]. The p(NIPAM) hydrogels synthesized in our study, using EGDM as a crosslinker, had significantly higher swelling degree than hydrogels cross-linked with *N*,*N′*-methylenebis(acrylamide) [43] or p(NIPAM) hydrogels obtained by UV polymerization [42].

The swelling profiles of p(NIPAM) hydrogels at 37 ± 1 °C and pH 2.2 and 7.4, are shown in Figure 4.

The hydrogel swelling degrees at 37 ± 1 °C (Figure 4) were multiple times lower than at 20 ± 1 °C (Figure 3). The swelling results at 37 ± 1 °C confirmed that the hydrogels were contracted at temperatures above VPTT, due to the stronger hydrophobic intramolecular interactions. From the aspect of crosslinker content, the hydrogels at 37 ± 1 °C behaved in the same way as hydrogels at 20 ± 1 °C: with increasing crosslinker content, the swelling degree of hydrogels decreased.

Kinetic parameters of swelling (diffusion exponent, *n*, swelling constant characteristic for a certain type of polymer network, *k*, and diffusion coefficient, *D*) of p(NIPAM) hydrogels at pH values of 2.2 and 7.4 are shown in Table 2.

The values of the diffusion exponent *n*, for synthesized homopolymeric hydrogels, are in the range of 0.51–0.73, which indicates anomalous transport, i.e., non-Fick diffusion of liquid into the gel (0.5 < *n* < 1) [38].

The sensitivity of p(NIPAM) hydrogels to the changes in temperature from 20 to 65 °C in the distilled water is given in Figure 5.

All tested p(NIPAM) hydrogels showed sensitivity to temperature changes. The swelling degree of p(NIPAM) hydrogels decreased with the temperature increase, which classifies them as negatively temperature-sensitive hydrogels. A sudden decrease in the swelling degree of hydrogels when the temperature changed from 33 °C to 34 °C represents the volume phase transition temperature [44]. A further increase in temperature led to an additional decrease in the swelling degree of hydrogels. The swelling degree decreased as expected with increasing crosslinker content, due to the increase of the crosslinking density which limits the mobility of the polymer chains [43].

### 3.3. p(NIPAM) Hydrogels as Carriers for Sulfanilamide

The successful synthesis of sulfanilamide was confirmed by applying the ^1^H-NMR method. The ^1^H-NMR spectrum of the synthesized sulfanilamide is shown in Figure 6.

In the ^1^H-NMR spectrum of sulfanilamide, two doublets originating from aromatic protons in positions H-2 and H-6, and H-3 and H-5 are observed at δ = 7.6549 and δ = 6.8594 ppm, respectively, which is in accordance with literature data [45,46]. Since deuterated water was used as a solvent, active hydrogen atoms from amino groups were replaced by deuterium, so there were no signals originating from the resonance of NH_2_ groups.

The reaction of hydrogels to temperature change and VPTT close to the physiological temperature of the human body classifies p(NIPAM) hydrogels into potential carriers of antimicrobial drugs. This is the reason why p(NIPAM) hydrogels have been examined as systems for modified sulfanilamide release.

#### 3.3.1. FTIR Analysis

FTIR spectra of sulfanilamide, homopolymeric p(NIPAM) hydrogel with 2.5 mol% of crosslinker, and the same hydrogel with incorporated sulfanilamide are shown in Figure 7.

In the FTIR spectrum of hydrogel with incorporated sulfanilamide, in the wavenumber range of 3500–3200 cm^−1^, four maxima originating from the valence vibrations of N-H bonds from sulfanilamide and homopolymer were observed. The maxima at 3385 cm^−1^ and 3320 cm^−1^ originating from asymmetric valence vibrations of N-H bonds from sulfonamide NH_2_ group and hydrogel, respectively, were shifted by 9 and 28 units toward higher wavenumbers, relative to the corresponding maxima in the spectra of sulfanilamide and hydrogel. The maximum at 3243 cm^−1^ which corresponds to symmetric valence vibrations of N-H bonds from the sulfonamide NH_2_ group was shifted for 25 units in relation to its position in the sulfanilamide spectrum (3268 cm^−1^). The change in intensity and position of these maxima indicates that these groups participate in the formation of hydrogen bonds between sulfanilamide and hydrogel. The maxima of amide band I, ν(C=O), at 1670 cm^−1^ and amide band II, δ(N-H), at 1577 cm^−1^, were shifted by 23 and 26 units, respectively, to higher wavenumbers, relative to their position in the spectrum of homopolymer. Shifting the maxima of amide bands and reducing their intensity indicates the participation of C=O and -NH groups in the formation of hydrogen bonds [47]. The appearance of characteristic bands in the spectrum of hydrogel with incorporated sulfanilamide at 1599 cm^−1^ and 1506 cm^−1^, originating from the valence vibrations of sulfanilamide aromatic ring C=C bond, two high intensity, sharp bands at 1313 cm^−1^ and 1154 cm^−1^, originating from the valence vibrations of SO_2_ group, and strong band at 1628 cm^−1^, originating from the deformation vibrations of sulfanilamide N-H groups confirm the presence of sulfanilamide inside the hydrogel. The changes in the intensity and position of characteristic bands of sulfanilamide and homopolymeric hydrogel indicate the incorporation of sulfanilamide into the hydrogel structure and the formation of hydrogen bonds between appropriate groups.

#### 3.3.2. XRD Analysis

The results of the XRD analysis are shown in Figure 8. The XRD of sulfanilamide is shown in Figure 8a. Several sharp reflection peaks indicated the crystalline structure of the synthesized sulfanilamide. All presented diffractograms of hydrogels (Figure 8b) contain two broad, unstructured peaks, whose intensity was slightly lower in the diffractogram of hydrogel with incorporated sulfanilamide. The secondary diffraction peak (2*θ*~8°) was the result of interaction between polymeric chains due to the presence of voluminous side groups, while the fundamental diffraction peak (2*θ*~20°) originated from intermolecular interactions of unbound atoms [48,49,50]. Wide, low-intensity diffraction peaks indicate a low degree of crystallinity of studied p(NIPAM) hydrogels, which is the reason why their structure has been described as amorphous in the literature [49,50]. This is favorable for their use as active substance carriers as it is easier to achieve uniform distribution of active substance molecules in amorphous than in partially crystalline polymer matrices. In the diffractogram of hydrogel with incorporated sulfanilamide, there were no peaks originating from sulfanilamide [35], which indicates that sulfanilamide was incorporated into the hydrogel.

#### 3.3.3. SEM Analysis

SEM micrographs of lyophilized p(NIPAM) hydrogels previously swollen to equilibrium, before and after sulfanilamide incorporation, are shown in Figure 9. SEM micrographs of lyophilized hydrogels before sulfanilamide incorporation (Figure 9a,b) showed a porous structure of the synthesized hydrogels, with pores of about 30 µm in diameter. In hydrogels with lower crosslinker content, the three-dimensional network was irregular with many interconnected pores. Hydrogels with higher crosslinker content contained pores of a more regular shape and thicker, channel-like walls, which gave a tubular microstructure to the homopolymeric hydrogels [51]. This kind of hydrogel structure facilitates the water passing through the hydrogel network, which is especially important in hydrogel application as carriers of active substances. By comparing SEM micrographs of hydrogels with incorporated sulfanilamide (Figure 9c,d) with micrographs of empty hydrogels, a clear difference in the appearance of surface hydrogel structures can be observed. The solubility of sulfanilamide in water is 7.5 mg/ml and in methanol about 27 mg/mL. Thus, a larger amount of sulfanilamide was introduced with methanol that can later be dissolved in the water in which the hydrogel swells. Therefore, the amount of sulfanilamide in the hydrogel is above the solubility limit, and molecular clusters are deposited in the pores of the gel. Irregularly distributed sulfanilamide crystals were observed, indicating that some part of the active substance was incorporated into the channels and pores of the hydrogel interior, while the other part remained on the surface.

By analyzing the results obtained using FTIR, XRD, and SEM methods, it can be concluded that sulfanilamide was successfully incorporated into the p(NIPAM) hydrogels.

#### 3.3.4. Sulfanilamide Incorporation into the p(NIPAM) Hydrogels

The loading efficiency of sulfanilamide into the polymeric network of p(NIPAM) hydrogels of different crosslinking degrees was determined in relation to the total available mass of sulfanilamide (*L_u_* = 500 mg/g_xerogel_). The values of sulfanilamide loading efficiency for p(NIPAM) hydrogels of different crosslinking degrees are shown in Table 3.

The presented results showed satisfactory efficiency of sulfanilamide incorporation in all p(NIPAM) hydrogels. The loading efficiency increased with the increase of hydrogel crosslinking degree, ranging from 87.36% for the hydrogel with 1 mol% of crosslinker to 95.29% for the hydrogel with 2.5 mol% of crosslinker.

#### 3.3.5. In Vitro Release of Sulfanilamide from p(NIPAM) Hydrogels

The sulfanilamide release from p(NIPAM) hydrogels at 37 ± 1 °C and pH 2.2 and 7.4 was monitored by the HPLC method. Under the defined conditions of the HPLC analysis, sulfanilamide showed a peak at the retention time of R_t_ = 2.259 min (Figure 10).

Release of sulfanilamide from p(NIPAM) hydrogels was monitored in vitro at 20 ± 1 and 37 ± 1 °C and pH 2.2 and 7.4, using the HPLC method. The results of these studies are shown in Figure 11 and Figure 12.

The obtained results showed that the highest amount of sulfanilamide was released from p(NIPAM) hydrogel with 1.5 mol% EGDM at pH 7.4 at 37 ± 1 °C. Under these conditions, 431.83 mg/g_xerogel_ was released, which represents 93.5% of incorporated sulfanilamide. A slightly lower amount of sulfanilamide was released from the same hydrogel in an acidic solution, 395.48 mg/g_xerogel_, or 85.6% of incorporated sulfanilamide, which is consistent with the results of the swelling analysis. It can be observed that with crosslinker content increase, the released amount of sulfanilamide decreases. Thus, only 336.24 mg/g_xerogel_ at pH 7.4 and 330.84 mg/g_xerogel_ at pH 2.2 of sulfanilamide were released from the hydrogel with 2.5 mol% EGDM, which represents 73.3% and 71.8% of incorporated sulfanilamide, respectively. It can be assumed that cross-linked structures interfere with hydrogel contraction at temperatures above VPTT, thereby slowing fluid diffusion from hydrogels [52,53]. Thus, higher crosslinker content leads to an increase of hydrogel network density, i.e., shortening branches between network nodes, higher fluid retention, and slower drug release from hydrogel, which is in accordance with the results of other authors [43].

The initial leaking occurred at the beginning of the process when the sulfanilamide molecules located on the surface of the swollen hydrogel, as well as in the surface pores, were washed away. Later, a stationary state of sulfanilamide diffusion from the deeper layers of the swollen hydrogel was established.

The release profile of sulfanilamide at 20 ± 1 °C and 37 ± 1 °C were similar, however, a lower amount of sulfanilamide was released from p(NIPAM) hydrogels at lower temperatures (lower amounts about 25–30%), presumably due to intermolecular interaction of sulfanilamide molecules, as well as the partial collapse of the hydrogel at a lower temperature, i.e., the decrease of the swelling degree and partial displacement of liquid from the gel. The collapse of the hydrogel allows a smaller contribution to the release of sulfanilamide at a lower temperature than the contribution of the collapse of the hydrogel at a higher temperature. But, at the same time, at a higher temperature, the contribution of diffusion is greater, and thus the overall effect is a greater release of sulfanilamide at a higher temperature. These findings are in accordance with the published results for similar hydrogels with different active substances, such as ibuprofen [33]. 

If the anomaly of the hydrogel with a 1% of crosslinker is excluded, it can be noticed that the amount of sulfanilamide released from the hydrogels decreased with an increase in the crosslinker concentration. On the one hand, due to the greater crosslinking degree, the pores of the hydrogel in the swollen state are smaller and the swelling degree is lower. Thus, the diffusion of sulfanilamide through the hydrogel to the external environment is difficult. On the other hand, the increased intensity of sulfanilamide interaction with the hydrogel of a higher crosslinking degree can additionally slow down the release of the drug from the hydrogel and contribute to the same phenomenon.

Lee and Lee synthesized hybrid hydrogels based on NIPAM and gelatin cross-linked with glutaraldehyde or genipin to investigate the effect of gelatin content and type of cross-linker on the incorporation and release of sulfanilamide from the hybrid hydrogels [17]. It is shown that with increasing gelatin content, the swelling degree of hydrogel decreases, whereby hydrogels cross-linked with glutaraldehyde has a higher degree of swelling than hydrogels cross-linked with genipin. The amount of incorporated sulfanilamide increases with increasing gelatin content, and this is more pronounced in hydrogels cross-linked with glutaraldehyde. On the other hand, the content of released sulfanilamide decreases with increasing gelatin content in the hybrid hydrogel. The amount of sulfanilamide released during 24 h ranges from 40 to 80% of incorporated sulfanilamide. Lee and Huang investigated the release of sulfanilamide, as a model drug, from a copolymer hydrogel based on NIPAM, glycidyl methacrylate (GMA) and *N*,*N*-dimethyl-acrylamide (DMA) [26], and showed that the amount of released of sulfanilamide increases with a decrease in the molar ratio of GMA/DMA, because the strength of the bond between the drug and the hydrogel depends on the content of GMA, whose epoxy groups can react with the amino groups of sulfanilamide and enable stronger binding of the drug to the hydrogel. The amount of released sulfanilamide varies in the range from 23.6 to 78.1% of incorporated sulfanilamide. Lee and Cheng investigated the effect of monomer content on sulfanilamide release and the properties of a biodegradable hydrogel based on NIPAM, acrylic acid, and poly(caprolactone)-diacrylate [27]. It was shown that the amount of incorporated sulfanilamide decreases with increasing acrylic acid content, due to the higher affinity of the hydrogel for water than for sulfanilamide. On the other hand, the amount of released sulfanilamide increases 4 times with an increase in the acrylic acid content from 0 to 50%, because the release of the drug depends primarily on the swelling degree, which also increases with an increase in the acrylic acid content, due to the stronger ability of the carboxyl groups of acrylic acid to bond water. The amount of released sulfanilamide varies in the range from 23.9 to 85.4% of incorporated sulfanilamide. Under the same conditions, a lower amount of sulfanilamide was released from each of the tested hydrogels compared to the synthesized p(NIPAM) hydrogels crosslinked with EGDM (73.3–93.5%).

In order to obtain a better insight into the sulfanilamide release kinetic, obtained experimental data were fitted using the appropriate mathematical models (Higuchi and Korsmeyer-Peppas) using DDSolver, and calculated parameters are shown in Table 4.

Higher values of the coefficient of determination (R^2^) and lower AIC values indicate that the Korsmeyer–Peppas model better describes the release of sulfanilamide from p(NIPAM) hydrogels. Values of the diffusion exponent *n* less than 0.5 indicate that the release of sulfanilamide from p(NIPAM) hydrogels of different crosslinking degrees is controlled by diffusion (“less to Fick’s” diffusion) at both pH values.

## 4. Conclusions

Homopolymeric p(NIPAM) hydrogels were successfully synthesized by free radical polymerization using EGDM as a crosslinker, structurally characterized using FTIR, XRD, and SEM methods, and examined as carriers for sulfanilamide. According to the results of the swelling study, the swelling of p(NIPAM) hydrogels depended on the temperature and crosslinker content while pH had no significant effect. Since the content of residual reactants was less than 2%, the synthesized p(NIPAM) hydrogels can be considered safe for use as active substance carriers. The FTIR analysis of the hydrogel with incorporated sulfanilamide indicated that the hydrogen bonds between polymeric chains and molecules of sulfanilamide are dominant. Wide, low-intensity diffraction peaks indicated a low degree of crystallinity of studied p(NIPAM) hydrogels, which is favorable for their use as active substance carriers. The sulfanilamide loading efficiency ranged from 87.36–95.29% and increased with the increase of crosslinker content. The release of sulfanilamide from the hydrogels was consistent with the swelling results: the increase of crosslinker content reduced the amount of sulfanilamide released. After 24 h, 73.3–93.5% of incorporated sulfanilamide was released from the hydrogels. The obtained results indicate the possibility of using thermosensitive homopolymeric p(NIPAM) hydrogels as carriers for sulfanilamide, for both local (vaginal, rectal) and systemic drug administration (peroral).

## Figures and Tables

**Figure 1 pharmaceutics-15-01749-f001:**
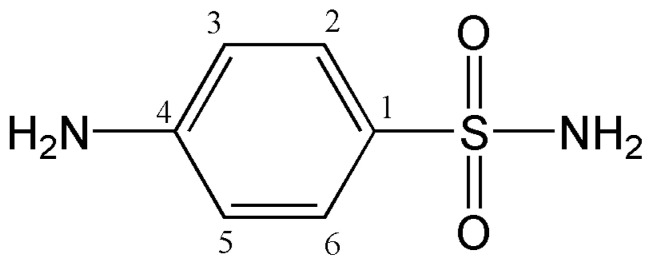
The chemical structure of sulfanilamide.

**Figure 2 pharmaceutics-15-01749-f002:**
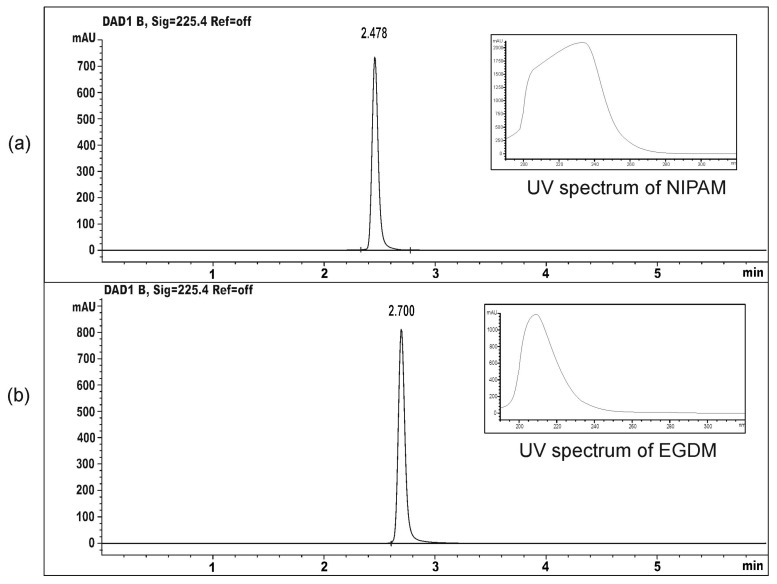
The HPLC chromatograms and UV spectra of: (**a**) monomer NIPAM and (**b**) crosslinker EGDM.

**Figure 3 pharmaceutics-15-01749-f003:**
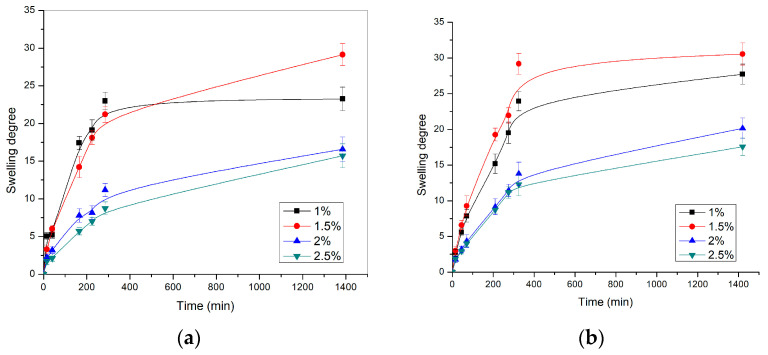
The swelling profiles of p(NIPAM) hydrogels at 20 ± 1 °C in the solution with the pH value of (**a**) 2.2 and (**b**) 7.4.

**Figure 4 pharmaceutics-15-01749-f004:**
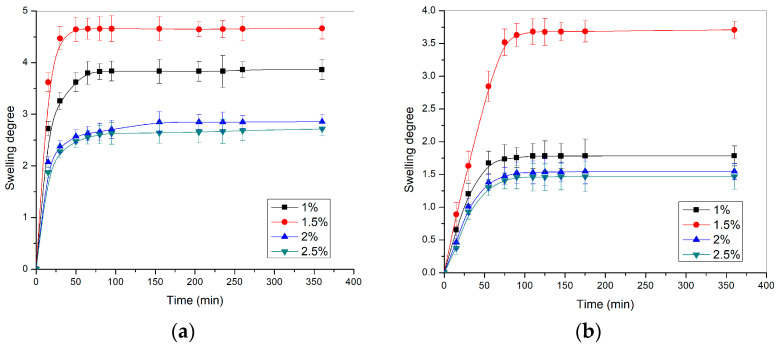
The swelling profiles of p(NIPAM) hydrogels at 37 ± 1 °C in the solution with the pH value of (**a**) 2.2 and (**b**) 7.4.

**Figure 5 pharmaceutics-15-01749-f005:**
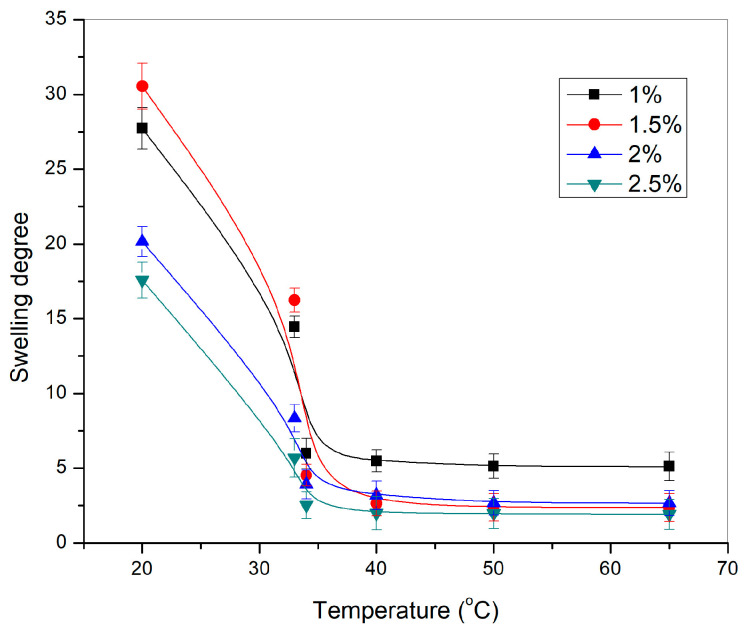
Dependence of the swelling degree of p(NIPAM) hydrogels, α, on the temperature.

**Figure 6 pharmaceutics-15-01749-f006:**
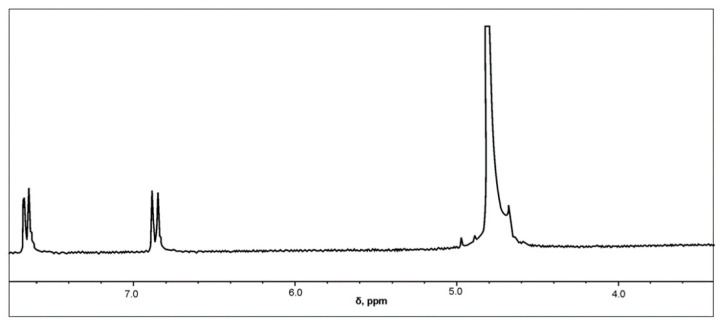
^1^H-NMR spectrum of sulfanilamide.

**Figure 7 pharmaceutics-15-01749-f007:**
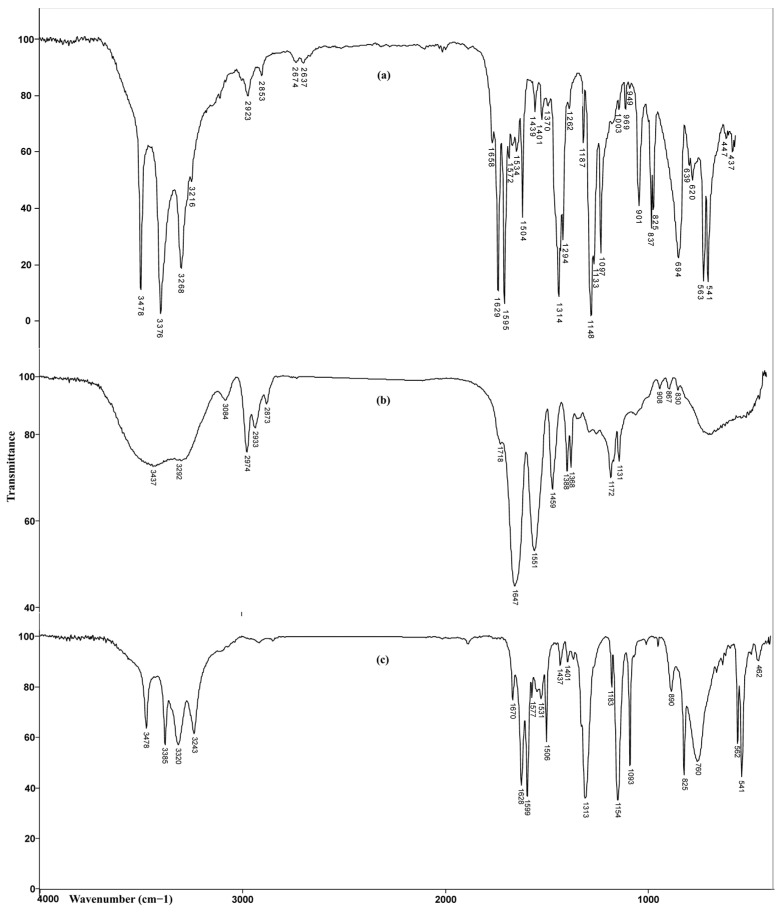
FTIR spectra of: (**a**) sulfanilamide, (**b**) p(NIPAM) hydrogel 2.5 mol%, and (**c**) p(NIPAM) hydrogel 2.5 mol% with incorporated sulfanilamide.

**Figure 8 pharmaceutics-15-01749-f008:**
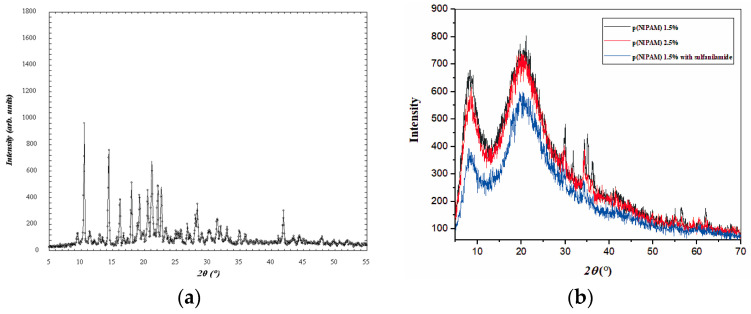
Diffractograms of (**a**) sulfanilamide, (**b**) p(NIPAM) with 1.5 and 2.5 mol% of crosslinker, and p(NIPAM) with 1.5 mol% crosslinker and incorporated sulfanilamide.

**Figure 9 pharmaceutics-15-01749-f009:**
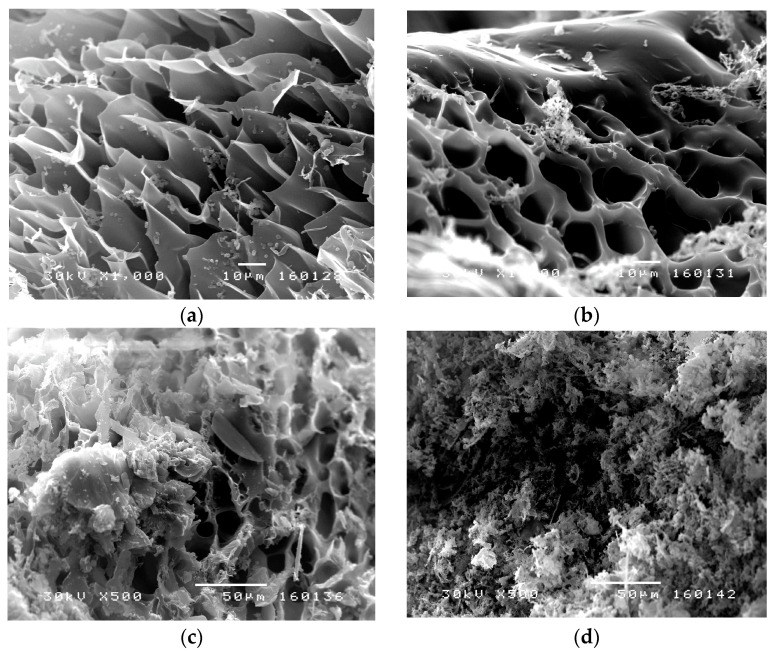
SEM micrographs of p(NIPAM) hydrogels: (**a**) 1.5 mol% (1000×), (**b**) 2.5 mol% (1000×), (**c**) 1.5 mol% with incorporated sulfanilamide (500×), and (**d**) 2.5 mol% with incorporated sulfanilamide (500×).

**Figure 10 pharmaceutics-15-01749-f010:**
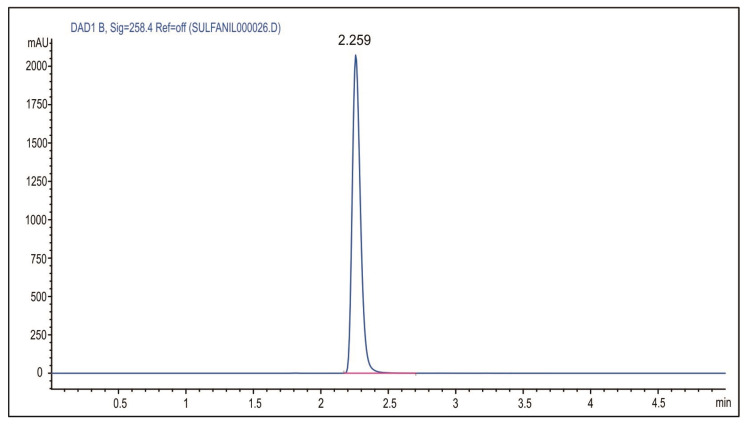
HPLC chromatogram of sulfanilamide.

**Figure 11 pharmaceutics-15-01749-f011:**
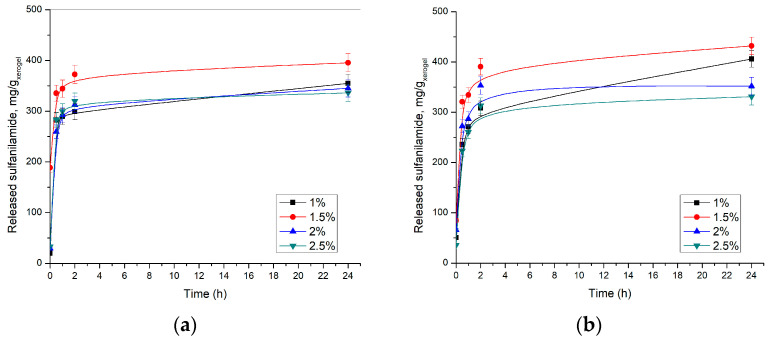
Released sulfanilamide content at 37 ± 1 °C from p(NIPAM) hydrogels at: (**a**) pH = 2.2 and (**b**) pH = 7.4.

**Figure 12 pharmaceutics-15-01749-f012:**
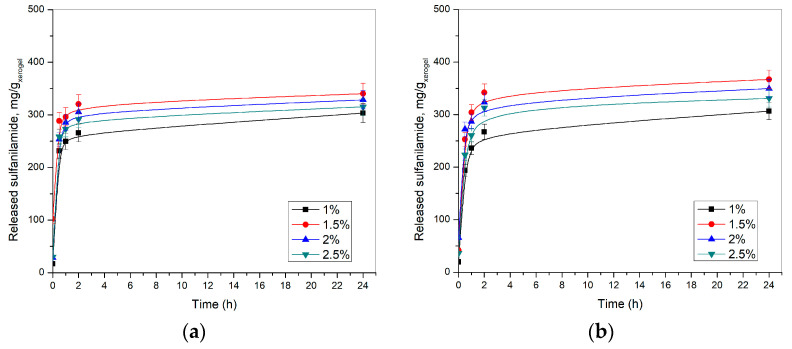
Released sulfanilamide content at 20 ± 1 °C from p(NIPAM) hydrogels at: (**a**) pH = 2.2 and (**b**) pH = 7.4.

**Table 1 pharmaceutics-15-01749-t001:** The content of residual reactants in synthesized p(NIPAM) hydrogels.

Sample	The Content of Residual Reactants in Sample, %			
	NIPAM, mg/g	NIPAM, %	EGDM, mg/g	EGDM, %
p(NIPAM) 1%	18.81 ± 1.32	1.91 ± 0.13	2.19 ± 0.13	1.73 ± 0.11
p(NIPAM) 1.5%	16.51 ± 0.91	1.69 ± 0.09	2.45 ± 0.19	1.88 ± 0.15
p(NIPAM) 2%	9.16 ± 0.37	0.95 ± 0.04	1.82 ± 0.10	1.86 ± 0.10
p(NIPAM) 2.5%	11.41 ± 0.40	1.19 ± 0.04	1.66 ± 0.05	1.29 ± 0.04

**Table 2 pharmaceutics-15-01749-t002:** Kinetic parameters of p(NIPAM) hydrogels swelling at pH 2.2 and 7.4 at 37 ± 1 °C.

Sample	pH = 2.2				pH = 7.4			
	*n*	*k* × 10^3^, min^1/*n*^	R^2^	*D* × 10^7^, cm^2^/min	*n*	*k* × 10^3^, min^1/*n*^	R^2^	*D* × 10^7^, cm^2^/min
p(NIPAM) 1%	0.54	50.38	0.999	49.82	0.66	16.46	0.998	5.32
p(NIPAM) 1.5%	0.61	21.73	0.999	9.27	0.73	13.68	0.999	3.67
p(NIPAM) 2%	0.51	31.83	0.986	19.89	0.62	15.26	0.991	4.57
p(NIPAM) 2.5%	0.57	20.62	0.964	8.34	0.60	18.75	0.975	6.90

*n*—diffusion exponent; *k*—the constant characteristic for a certain type of polymer network (min^1/n^); *D*—diffusion coefficient (cm^2^/min).

**Table 3 pharmaceutics-15-01749-t003:** The masses of xerogels and incorporated sulfanilamide (*L_g_*) and loading efficiency (*η*).

Sample	Mass of Xerogel, g	*L_g_*, mg/g_xerogel_	*η*_sulfanilamide_, %
p(NIPAM) 1%	0.0473	461.74	87.36
p(NIPAM) 1.5%	0.0509	450.55	91.73
p(NIPAM) 2%	0.0489	479.91	93.87
p(NIPAM) 2.5%	0.0509	468.05	95.29

*L_g_*—mass of incorporated sulfanilamide (mg/g_xerogel_); *η*—loading efficiency.

**Table 4 pharmaceutics-15-01749-t004:** The kinetic of sulfanilamide release from p(NIPAM) hydrogels.

Kinetic Model	Parameter	pH = 2.2				pH = 7.4			
		1%	1.5%	2%	2.5%	1%	1.5%	2%	2.5%
Higuchi *F* = *k_H_* ∙ *t*^1/2^	*k_H_*	7.290	9.075	7.055	7.507	6.974	9.116	7.507	6.786
	R^2^	0.714	0.488	0.794	0.736	0.782	0.634	0.742	0.882
	AIC	28.418	30.737	26.540	27.944	25.975	29.600	26.974	23.841
Korsmeyer–Peppas *F* = *k_KP_* ∙ *t^n^*	*k_KP_*	53.229	56.767	34.855	44.929	26.621	42.202	28.375	20.528
	*n*	0.041	0.077	0.132	0.087	0.192	0.147	0.194	0.246
	R^2^	0.993	0.848	0.983	0.980	0.935	0.865	0.898	0.970
	AIC	15.681	27.868	18.652	19.635	23.147	27.601	25.249	20.314

*k_H_*: the constant of release at Higuchi model; R^2^: coefficient of determination; AIC: Akaike’s information criterion; *F*: the fraction of released drug in the function of time *t*; *k_KP_*: the constant of release at Korsmeyer-Peppas model that takes into account structural and geometrical characteristics of the dosage form; *n*: the exponent of release/diffusion [54,55].

## Data Availability

Not applicable.

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
