# Peer review of "Modified Sulfanilamide Release from Intelligent Poly(*N*-isopropylacrylamide) Hydrogels"

_pharmaceutics, 2023, doi:10.3390/pharmaceutics15061749_

Round 1
Reviewer 1 Report
The paper present the synthesis and characterization of a well-known hydrogel based on poly(N-isopropylacrylamide) for the delivery of sulfanilamide. Some major problems can be observed:
Different papers present the synthesis and characterization of hydrogels/microgels based on p(NIPAM) cross-linked with EGDME [https://doi.org/10.15671/hjbc.719698 , DOI:10.1007/s00396-004-1160-x, https://doi.org/10.2478/s11696-010-0065-z , including ref 29]. The novelty of the present paper is represented only by the used of this hydrogels for the release of sulfanilamide?
What is the purpose of using p(NIPAM) in the presented drug release system, the hydrophobic interaction between the polymer and sulfanilamide?
If the ability of the hydrogels to respond to temperature was taken into consideration, the release of the drug at different temperatures must be study (below and above the VPTT).
What concrete applications are taken into account for these hydrogels loaded with sulfanilamide?
In the title it is used „modified sulfanilamide release”, but simple sulfanilamide was used in this work.
There are also other question that need to be clarified:
Pag 2, lines 81-82: Reference [16] deal with double network hydrogels based on poly(2-acrylamido-2-methylpropane sulfonic acid) and PNIPAm, so it cannot be said that “The VPTT of p(NIPAM) hydrogel is 33-35 °C” according to [16]. Because the hydrogels are too big (compared to microgels) the decrease of their swelling with the temperature require a long time. For microgels, the VPTT can be easier measured. The VPTT for hydrogels/microgels from p(NIPAM) homopolymer is measured in the literature : http://dx.doi.org/10.1063/1.447548 , doi:10.3390/polym11040620 , https://doi.org/10.1002/macp.201700364., https://doi.org/10.1007/s00396-020-04632-5, DOI: 10.1002/macp.201100340 , or other papers. The same comment is for page 9, lines 331-332.
In the chapter “2.3. Hydrogels synthesis”, the concentration of the NIPAM used for the synthesis should be mentioned. It cannot be understood if the 1, 1.5, 2 and 2.5 mol% crosslinker is the ratio between EGDM and NIPAM, or the concentration of EGDM in the reaction mixture. The notation of the samples (p(NIPAM) 1%,…, p(NIPAM) 2.5%) should be explained in this chapter.
If all the hydrogel samples are subject to lyophilization, the chapter “2.4. Lyophilization of hydrogels” can be presented in the chapter “Hydrogels synthesis”.
In the chapter “2.6. Swelling of hydrogels”, “the dry hydrogel samples (xerogels)” are the samples obtained after lyophilization? It shoul de mentioned if the pH =2.2 or 7.4 were obtained from water by the addition of acid/base, or from buffers.
The equation (4) was not introduced/ demonstrated by/in reference [30] from 2023, where the authors study the release of curcumin (inappropriate self-citation). This equation was introduced by Ritger and Peppas [https://doi.org/10.1016/0168-3659(87)90034-4 or https://doi.org/10.1016/0168-3659(87)90035-6].
The “Sulfanilamide incorporation into the hydrogels” must be presented before the chapter 2.7.1 where it is written that “FTIR spectra of sulfanilamide, synthesized xerogels and xerogels with incorporated sulfanilamide were recorded….”
In the chapter 2.7.5 it is written that “The swollen hydrogels with incorporated sulfanilamide….”. It is referred to the hydrogels swollen in methanol, or in water?
If the experiments regarding the swelling of the hydrogels were performed in triplicate, the results in the figures should be presented with error bar (standard deviation).
The strange fact that the hydrogel with 1.5% crosslinking have a higher swelling compared with the hydrogel with 1% crosslinking was explained “as sufficient crosslinking density of hydrogels was not reached due to the low crosslinker concentration”. Please elaborate this explanation.
At pag 8, lines 311-313 it is written that “the hydrogels at 37±1 °C behaved in the same way as hydrogels at 20±1°C: with increasing crosslinker content, the swelling degree of hydrogels decreased. ”, but the hydrogel with 1.5% crosslinking does not respect this rule.
From the manuscript it can’t be understand if the parameters from Table 2 were obtained from the swelling at 20 °C or at 37 °C.
How were performed the measurement of the swelling degree from the Figure 5? The same hydrogel was immersed in water at 20 °C, then the temperature was increased to 33 °C, or different samples were swollen in water at different temperatures? What was the pH used for these experiments?
In Figure 4, the swelling degree for p(NIPAM)1.5% hydrogels at 37 °C was higher compared to p(NIPAM)1%, but in Figure 5 it seems that the swelling degree at was higher for p(NIPAM)1% compared to p(NIPAM)1.5% at 37 °C or 40 °C.
Pag 13, line 423: Reference [25] state that amino groups from trypsin (or from sulfanilamide in this case) can react with the epoxy group of glycidyl methacrylate, but EGDM does not contain epoxy groups.
-
Author Response
Comments and Suggestions for Authors
The paper present the synthesis and characterization of a well-known hydrogel based on poly(N-isopropylacrylamide) for the delivery of sulfanilamide. Some major problems can be observed:
Different papers present the synthesis and characterization of hydrogels/microgels based on p(NIPAM) cross-linked with EGDME [https://doi.org/10.15671/hjbc.719698, DOI:10.1007/s00396-004-1160-x, https://doi.org/10.2478/s11696-010-0065-z , including ref 29]. The novelty of the present paper is represented only by the used of this hydrogels for the release of sulfanilamide?
Answer:
Poly(N-isopropylacrylamide) hydrogel is not an innovative system. It was used in many studies as a carrier of different model substances, as you mentioned. In this work, the formulation of the mentioned carrier was prepared and tested with sulfanilamide as a model substance. This research is important because sulfanilamide is a significant structural element of the entire sulfonamide class of drugs, responsible for the antibacterial activity of this drug class. In direct and indirect contact with mouse fibroblasts, p(NIPAM) hydrogel does not reduce the viability of fibroblast cells and thus shows minimal cytotoxicity (Samantha A. Meenach, A. Ashley Anderson, Mehul Suthar, Kimberly W. Anderson, J. Zach Hilt, Biocompatibility analysis of magnetic hydrogel nanocomposites based on poly(N-isopropylacrylamide) and iron oxide), Journal of Biomedical Materials Research, 17 December 2008 https://doi.org/10.1002/jbm.a.32322). For these reasons, a combination of p(NIPAM) hydrogel and sulfanilamide was chosen and examined.
What is the purpose of using p(NIPAM) in the presented drug release system, the hydrophobic interaction between the polymer and sulfanilamide?
Answer:
Poly(N-isopropylacrylamide) hydrogel is a thermosensitive hydrogel, so it is suitable as a carrier for drugs used in the treatment of infections accompanied by inflammation, and from that aspect it was chosen as a carrier for sulfanilamide. We considered that this combination of a carrier and a drug is interesting and that some initial research should be carried out that would indicate the importance of such a formulation, which should certainly be further examined from different aspects.
If the ability of the hydrogels to respond to temperature was taken into consideration, the release of the drug at different temperatures must be study (below and above the VPTT).
Answer:
The homopolymer p(NIPAM) has a lower critical solution temperature (LCST) of 32 °C. The manuscript shows the release of sulfanilamide at 37 °C (which is above the LCST). Also, according to the suggestion of the reviewer, the release results at 20 °C (which is below the LCST) were added.
What concrete applications are taken into account for these hydrogels loaded with sulfanilamide?
Answer:
The proposed system can be used for local (vaginal) or peroral administration of the drug.
In the title it is used „modified sulfanilamide release”, but simple sulfanilamide was used in this work.
Answer:
We think there is a misunderstanding here. It is not the sulfanilamide molecule that has been modified, rather this is a common term for changed release profile of active substance from a hydrogel formulation compared to some classical pharmaceutical formulations.
There are also other question that need to be clarified:
Pag 2, lines 81-82: Reference [16] deal with double network hydrogels based on poly(2-acrylamido-2-methylpropane sulfonic acid) and PNIPAm, so it cannot be said that “The VPTT of p(NIPAM) hydrogel is 33-35 °C” according to [16]. Because the hydrogels are too big (compared to microgels) the decrease of their swelling with the temperature require a long time. For microgels, the VPTT can be easier measured. The VPTT for hydrogels/microgels from p(NIPAM) homopolymer is measured in the literature: http://dx.doi.org/10.1063/1.447548, doi:10.3390/polym11040620, https://doi.org/10.1002/macp.201700364, https://doi.org/10.1007/s00396-020-04632-5, DOI: 10.1002/macp.201100340 , or other papers. The same comment is for page 9, lines 331-332.
Answer:
The authors agree with the reviewer's suggestion and this statement will be removed from the manuscript. The authors thank the reviewer.
In the chapter “2.3. Hydrogels synthesis”, the concentration of the NIPAM used for the synthesis should be mentioned. It cannot be understood if the 1, 1.5, 2 and 2.5 mol% crosslinker is the ratio between EGDM and NIPAM, or the concentration of EGDM in the reaction mixture. The notation of the samples (p(NIPAM) 1%,…, p(NIPAM) 2.5%) should be explained in this chapter.
Answer:
The data refers to the amount of crosslinker in relation to the amount of monomer. Thanks for the suggestion, it has been corrected in the manuscript.
If all the hydrogel samples are subject to lyophilization, the chapter “2.4. Lyophilization of hydrogels” can be presented in the chapter “Hydrogels synthesis”.
Answer:
Thank you for the suggestion, section 2.4. is appended to section 2.3. and now represents a single section.
In the chapter “2.6. Swelling of hydrogels”, “the dry hydrogel samples (xerogels)” are the samples obtained after lyophilization? It shoul de mentioned if the pH =2.2 or 7.4 were obtained from water by the addition of acid/base, or from buffers.
Answer:
Aqueous solutions were prepared by adding hydrochloric acid and sodium hydroxide until the desired pH values were reached.
The equation (4) was not introduced/ demonstrated by/in reference [30] from 2023, where the authors study the release of curcumin (inappropriate self-citation). This equation was introduced by Ritger and Peppas [https://doi.org/10.1016/0168-3659(87)90034-4 or https://doi.org/10.1016/0168-3659(87)90035-6].
Answer:
Thanks for the suggestion; the literature reference has been replaced according to your recommendation, Ritger and Pepas [Ritger, P.L.; Peppas, N.A. A simple equation for description of solute release II. Fickian and anomalous release from swellable devices. J. Control. Release 1987, 5, 37–42].
The “Sulfanilamide incorporation into the hydrogels” must be presented before the chapter 2.7.1 where it is written that “FTIR spectra of sulfanilamide, synthesized xerogels and xerogels with incorporated sulfanilamide were recorded….”
Answer:
Thanks for the suggestion; the section 2.7.4. Sulfanilamide incorporation into the hydrogels is now presented before section 2.7.
In the chapter 2.7.5 it is written that “The swollen hydrogels with incorporated sulfanilamide…”. It is referred to the hydrogels swollen in methanol, or in water?
Answer:
Thanks for the suggestion. A clarification has been added to the manuscript. Hydrogel with sulfanilamide was dried from methanol and then was in contact with only the amount of water that it can absorb according to the degree of swelling in order to avoid premature release of sulfanilamide. After that, the release of sulfanilamide in the swollen state was monitored.
If the experiments regarding the swelling of the hydrogels were performed in triplicate, the results in the figures should be presented with error bar (standard deviation).
Answer:
Error bars were added to the figures. The authors are grateful for the suggestion.
The strange fact that the hydrogel with 1.5% crosslinking have a higher swelling compared with the hydrogel with 1% crosslinking was explained “as sufficient crosslinking density of hydrogels was not reached due to the low crosslinker concentration”. Please elaborate this explanation.
Answer:
The hydrogel synthesized with 1% of crosslinker showed that the created polymer network is not optimal for reaching a high swelling degree. The assumption is that some parts of the polymer network have too long branches and too few network nodes due to the low concentration of crosslinker. These parts of the polymer network partially dissolve and do not contribute to the swelling degree. Hydrogel with 1.5% crosslinker has a higher degree of swelling due to a favorable ratio of branch lengths and network nodes. With the further increase in the concentration of crosslinker, the branches are shortened and the concentration of network nodes increases, which reduces the possibility of expansion of the network structure, thus decreasing the swelling degree.
At pag 8, lines 311-313 it is written that “the hydrogels at 37±1 °C behaved in the same way as hydrogels at 20±1°C: with increasing crosslinker content, the swelling degree of hydrogels decreased. ”, but the hydrogel with 1.5% crosslinking does not respect this rule.
Answer:
The explanation is the same as for the previous statement.
From the manuscript it can’t be understand if the parameters from Table 2 were obtained from the swelling at 20 °C or at 37 °C.
Answer:
Thanks for the suggestion. In the title of Table 2, the temperature of 20 °C was added.
How were performed the measurement of the swelling degree from the Figure 5? The same hydrogel was immersed in water at 20 °C, then the temperature was increased to 33 °C, or different samples were swollen in water at different temperatures? What was the pH used for these experiments?
Answer:
Each experiment was conducted so that the swellings were monitored in distilled water at the appropriate temperatures for different hydrogel samples. The temperature of one hydrogel sample was not increased.
In Figure 4, the swelling degree for p(NIPAM)1.5% hydrogels at 37 °C was higher compared to p(NIPAM)1%, but in Figure 5 it seems that the swelling degree at was higher for p(NIPAM)1% compared to p(NIPAM)1.5% at 37 °C or 40 °C.
Answer:
Figure 5 shows that the swelling degree of the hydrogel with 1% of crosslinker had lower values than the hydrogel with 1.5% crosslinker at temperatures up to 33 °C. With increasing temperature, this position has changed. The hydrogel with 1.5% of crosslinker had the largest change in swelling degree with increasing temperature, which may indicate its optimal network structure in terms of swelling.
Pag 13, line 423: Reference [25] state that amino groups from trypsin (or from sulfanilamide in this case) can react with the epoxy group of glycidyl methacrylate, but EGDM does not contain epoxy groups.
Answer:
Here, the authors wanted to indicate the possible participation of the amino group in the formation of hydrogen bonds with hydrogels, which has already been described in the literature. However, we will remove this statement from the manuscript to avoid confusion. The authors thank the reviewer.
Comments on the Quality of English Language
-
Submission Date
15 April 2023
Date of this review
26 Apr 2023 08:48:29

Reviewer 2 Report
This work deals with the synthesis and characterization of homopolymer poly(N-isopropylacrylamide) p(NIPAM) hydrogels cross-linked with ethylene glycol dimethacrylate (EGDM) as well as the evaluation of such hydrogels for the controlled sulfanilamide release. The following issues are requested to the authors.
1. Introduction section: Authors commented that previous works have reported the preparation and evaluation of sulfanilamide-polymer carriers for the controlled release of this drug. The motivation and the novelty of the present research should be further highlighted. It is not enough to say that it is the first time that the p(NIPAM)/ sulfanilamide system is studied.
2. The research core of this work was the development of PNIPAAm-based hydrogels for controlled drug delivery. To balance the scientific viewpoint and attract more attention from audiences, the authors are highly recommended to consider the inclusion of the recent relevant papers, such as: https://doi.org/10.3390/pharmaceutics13081284
3. Error bars should be included in figures 3, 4, and 5.
4. Table 2. Authors should indicate the temperature of the swelling data. How was the diffusion coefficient determined?
5. The quality of the FTIR figure should be improved. Some of the peak labels are illegible and it is necessary to homogenize the thickness of the lines of the spectra.
6. How do authors explain the fact that the drug loading efficiency increases with increasing the crosslinking degree of hydrogel?
7. Figure 10. The drug release profile showed a burst release within the first hour and the equilibrium condition was reached at short times. Were the drug release experiments done from relaxed or dry hydrogels?
8. Authors should discuss and compare the sulfanilamide release results with similar literature reports. What are the benefits of using p(NIPAM) hydrogels in sulfonamide release compared to other reported systems?
9. How could the proposed system be used in practice?
Minor editing of English language is required.
Author Response
Comments and Suggestions for Authors
This work deals with the synthesis and characterization of homopolymer poly(N-isopropylacrylamide) p(NIPAM) hydrogels cross-linked with ethylene glycol dimethacrylate (EGDM) as well as the evaluation of such hydrogels for the controlled sulfanilamide release. The following issues are requested to the authors.
- Introduction section: Authors commented that previous works have reported the preparation and evaluation of sulfanilamide-polymer carriers for the controlled release of this drug. The motivation and the novelty of the present research should be further highlighted. It is not enough to say that it is the first time that the p(NIPAM)/ sulfanilamide system is studied.
Answer:
In direct and indirect contact with mouse fibroblasts, p(NIPAM) hydrogel does not reduce the viability of fibroblast cells and thus shows minimal cytotoxicity (Samantha A. Meenach, A. Ashley Anderson, Mehul Suthar, Kimberly W. Anderson, J. Zach Hilt, Biocompatibility analysis of magnetic hydrogel nanocomposites based on poly(N-isopropylacrylamide) and iron oxide), Journal of Biomedical Materials Research, 17 December 2008 https://doi.org/10.1002/jbm.a.32322). This fact, as well as the fact that sulfanilamide belongs to an important group of antimicrobial agents and that no research on the p(NIPAM)/sulfanilamide system has been published so far, was the motivation for this study. In the literature, there are formulations of sulfanilamide with other carriers, but not with p(NIPAM).
- The research core of this work was the development of PNIPAAm-based hydrogels for controlled drug delivery. To balance the scientific viewpoint and attract more attention from audiences, the authors are highly recommended to consider the inclusion of the recent relevant papers, such as: https://doi.org/10.3390/pharmaceutics13081284
Answer:
The authors are grateful for the suggestion. The proposed reference is included into the analysis and discussion of the results.
- Error bars should be included in figures 3, 4, and 5.
Answer:
Error bars were added to the figures. The authors are grateful for the suggestion.
- Table 2. Authors should indicate the temperature of the swelling data. How was the diffusion coefficient determined?
Answer:
The temperature for Table 2 has been added to the manuscript. The diffusion coefficient was determined using equations 5 and 6 added to the manuscript.
- The quality of the FTIR figure should be improved. Some of the peak labels are illegible and it is necessary to homogenize the thickness of the lines of the spectra.
Answer:
We improved the quality of the figures, including FTIR figure. All of the figures are in resolution of 1200 dpi.
- How do authors explain the fact that the drug loading efficiency increases with increasing the crosslinking degree of hydrogel?
Answer:
The authors cannot explain this phenomenon, probably the absorption mechanism of the methanol solution of sulfanilamide by lyophilized p(NIPAM) hydrogels changes.
- Figure 10. The drug release profile showed a burst release within the first hour and the equilibrium condition was reached at short times. Were the drug release experiments done from relaxed or dry hydrogels?
Answer:
The experiments were performed by releasing sulfanilamide from the dry gel. Presumably, the initial leaking occurs at the beginning of the process when the sulfanilamide molecules located on the surface of the dry gel, as well as in the surface pores, are washed away. Later, a stationary state of sulfanilamide molecules diffusion from the deeper layers of the swollen hydrogel is established.
This comment was included in the discussion section of the manuscript.
- Authors should discuss and compare the sulfanilamiderelease results with similar literature reports. What are the benefits of using p(NIPAM) hydrogels in sulfonamide release compared to other reported systems?
Answer:
In the available literature, the authors did not find similar results that can be compared with the results in this manuscript. Studies of sulfanilamide with cyclodextrins and alginate gels have been published, but not with thermosensitive hydrogels. The reason why p(NIPAM) was chosen is given in the answer to suggestion 1.
- How could the proposed system be used in practice?
Answer:
The proposed system can be used for local (vaginal) or peroral administration of the drug.
Comments on the Quality of English Language
Minor editing of English language is required.
Answer:
The manuscript was revised by a native English speaker.
Submission Date
15 April 2023
Date of this review
05 May 2023 17:21:51

Reviewer 3 Report
The manuscript entitled "Modified Sulfanilamide release from intelligent poly(N-isopropylacrylamide) hydrogels" submitted by Gajic and coworkers to Pharmaceutics describes the use of hydrogels based on N-isopropylacrylamide and ethylene glycol dimethacrylate at different crosslinking degress as materials for drug delivery. Concretely, the authors study the loading and release of the antibiotic sulfanilamide.
The study is interesting and relevant; the objective is clear and so the title and the abstract are appropriate. The manuscript is clearly written and reads well. The reviewer recommends for its publication in Pharmaceutics after addressing minor comments that can benefit the publication. They are given below.
1. In page 3, line 101, the authors refer to different hydrogels previously tested as carriers of sulfanilamide. Please, add a table with the results obtained in these studies and compare your data with them during the discussion of your results.
2. Page 3, line 134. Please, justify why a temperature ramp is used for the polymerization of the hydrogels instead of carrying out the polymerization at constant temperature. It can be done just by adding new references that support it.
3. Page 5, 200 and page 12, line 410. Please, explain better the experiment to calculate the sulfanilamide incorporation. Do you dry the hydrogels after the loading and then wheith them? The Lg data in the table have too many decimal numbers and the experiment seems to have a different level of uncertainty.
4. Page 7, line 261. The sentence "After the extraction and drying, it was not possible to detect residual amount of reactants in the synthesized hydrogels" sounds too vague. If it was not possible to carry out the experiments, the authors should try to repeat it or justify why it was not possible. If the experiments were carried out but no residuals amount of reactants were detected, please, specify it correctly.
5. Page 13, line 425. The authors suggest that the release of sulfanilamide could be triggered by temperature. However, although the swelling has been studied at different temperature, the release of the drug is only studied at 37.0. Please, compare your results at 37º with the same experiment carried out at lower temperature to justified that the release depends on the temperature.
6. Page 13, Line 437. The authors suggest in line 421 that the EGDM in the hydrogels interacts with the sulfanilamide. Therefore, could the lower release of the sulfanilamide from high crosslinked hydrogels also be related with a higher interaction of the drug with the hydrogel? Please, consider this possibility in the discussion of the results.
Author Response
Comments and Suggestions for Authors
The manuscript entitled "Modified Sulfanilamide release from intelligent poly(N-isopropylacrylamide) hydrogels" submitted by Gajic and coworkers to Pharmaceutics describes the use of hydrogels based on N-isopropylacrylamide and ethylene glycol dimethacrylate at different crosslinking degress as materials for drug delivery. Concretely, the authors study the loading and release of the antibiotic sulfanilamide.
The study is interesting and relevant; the objective is clear and so the title and the abstract are appropriate. The manuscript is clearly written and reads well. The reviewer recommends for its publication in Pharmaceutics after addressing minor comments that can benefit the publication. They are given below.
- In page 3, line 101, the authors refer to different hydrogels previously tested as carriers of sulfanilamide. Please, add a table with the results obtained in these studies and compare your data with them during the discussion of your results.
Answer:
The authors added a discussion and comparison with the results of other authors in the discussion section after Figure 12 and before Table 4.
- Page 3, line 134. Please, justify why a temperature ramp is used for the polymerization of the hydrogels instead of carrying out the polymerization at constant temperature. It can be done just by adding new references that support it.
Answer:
Some of the authors of this paper have been studying the synthesis of polymers based on N-isopropylacrylamide since 2014. A procedure with temperature ramps was developed in order to achieve the highest possible degree of conversion and the minimum amount of residual monomers and initiators at the end of the polymerization process. If the highest temperature was used at the beginning of the polymerization process, a high rate of initiator decomposition (the initiator's half-time of decomposition is shortened) and a large amount of radicals as active centers for polymerization and formation of a cross-linked polymer would be obtained. This could intensify the radical recombination side reactions, which would reduce the initiator's efficiency. The authors have previously achieved good results in terms of the quality of hydrogels with this synthesis procedure.
- Page 5, 200 and page 12, line 410. Please, explain better the experiment to calculate the sulfanilamide incorporation. Do you dry the hydrogels after the loading and then wheith them? The Lg data in the table have too many decimal numbers and the experiment seems to have a different level of uncertainty.
Answer:
Yes, after swelling of lyophilized hydrogels in a methanolic solution of sulfanilamide, the swollen hydrogels are lyophilized again. After that, the dry gels with sulfanilamide are brought into contact with only the amount of water that they can absorb according to the degree of swelling so that premature release of sulfanilamide does not occur. Then used for examination of sulfanilamide release from this formulation. The largest part of sulfanilamide enters the pores of the gel, but a smaller part remains in the surface pores. From those surface pores, sulfanilamide molecules will first be released into the surrounding environment, and this effect is known as the leaking effect.
- Page 7, line 261. The sentence "After the extraction and drying, it was not possible to detect residual amount of reactants in the synthesized hydrogels" sounds too vague. If it was not possible to carry out the experiments, the authors should try to repeat it or justify why it was not possible. If the experiments were carried out but no residuals amount of reactants were detected, please, specify it correctly.
Answer:
Table 1 shows the results of determining the concentration of residual monomers in hydrogels before extraction of hydrogels with methanol. Application of temperature ramps during synthesis and methanol extraction after hydrogel synthesis practically reduced the amount of residual monomers to the extent that they were no longer detected in hydrogels. That was the goal, considering that the hydrogel formulations with sulfanilamide is intended for use in pharmacy. Polymers based on N-isopropylacrylamide are not cytotoxic (Samantha A. Meenach, A. Ashley Anderson, Mehul Suthar, Kimberly W. Anderson, J. Zach Hilt, Biocompatibility analysis of magnetic hydrogel nanocomposites based on poly(N-isopropylacrylamide) and iron oxide), Journal of Biomedical Materials Research, 17 December 2008 https://doi.org/10.1002/jbm.a.32322) but there is no information about cytotoxicity of monomers that can be found in the polymer as residuals. The authors chose the hydrogel synthesis and processing procedure to eliminate any risk of cytotoxicity of the p(NIPAM)/sulfanilamide formulation.
The sentence is rewritten in the following manner:
“After the extraction and drying, no residual reactants were detected in the synthesized hydrogels.”
- Page 13, line 425. The authors suggest that the release of sulfanilamide could be triggered by temperature. However, although the swelling has been studied at different temperature, the release of the drug is only studied at 37.0. Please, compare your results at 37º with the same experiment carried out at lower temperature to justified that the release depends on the temperature.
Answer:
Thanks for the suggestion, the analysis was also performed at 20°C and shown in the manuscript.
- Page 13, Line 437. The authors suggest in line 421 that the EGDM in the hydrogels interacts with the sulfanilamide. Therefore, could the lower release of the sulfanilamide from high crosslinked hydrogels also be related with a higher interaction of the drug with the hydrogel? Please, consider this possibility in the discussion of the results.
Answer:
This is a very interesting observation by the reviewer and the authors are grateful for this suggestion. On the one hand, due to greater cross-linking degree of the hydrogel, the pores of the hydrogel in the swollen state are smaller because the degree of swelling is also lower and thus the diffusion of sulfanilamide through the hydrogel to the external environment is difficult. On the other hand, the increased intensity of sulfanilamide interaction with the hydrogel of higher crosslinking can additionally slow down the release of the drug from the hydrogel and contribute to the same phenomenon.
Submission Date
15 April 2023
Date of this review
05 May 2023 12:33:30

Reviewer 4 Report
Although the general idea of the work is interesting, the description of the experiments and the experimental approach need to be described and justified more in detail.
Some experiments are redundant: since the starting polymer has no ionizable functional groups it is clear that the swelling cannot depend on the pH.
Below specific comments:
· the correlation between hydrogel VPTT and the actual drug delivery advantage is unclear.
· the drug spectrum must be reported in the diffractogram
· since the drug was synthesized de novo, a more in-depth characterization must be reported e.g. 1H-NMR
· line 143: what do the authors mean by swollen quickly?
· the authors demonstrated that the purification procedure is effective in eliminating uncrosslinked monomers. however, no analysis has been conducted to demonstrate that there are no traces of methanol.
· the procedure for drug incorporation needs to be described in more detail. have they been washed? if yes, how many and with which solvent.
How was the methanol eliminated?
· Lines 210-211: authors should specify the medium.
· Swelling results lack of statistical analysis. Also, the reason why 1.5% hydrogel shows higher swelling is poorly explained.
· Since the main stimulus that influences the hydrogel swelling behavior is the temperature, release experiment must be conducted at different temperatures (25 and 37°C). This is crucial to demonstrate that entire rationale of the work.
Author Response
Comments and Suggestions for Authors
Although the general idea of the work is interesting, the description of the experiments and the experimental approach need to be described and justified more in detail.
Some experiments are redundant: since the starting polymer has no ionizable functional groups it is clear that the swelling cannot depend on the pH.
Below specific comments:
- the correlation between hydrogel VPTT and the actual drug delivery advantage is unclear.
Answer:
The paragraph [lines 80-84] was supplemented with the following sentence:
“At the physiological body temperature that is higher than the volume phase transition temperature of p(NIPAM) hydrogels, the intermolecular interactions between sulfanilamide and side groups of the polymer matrix get broken and contraction of the polymer matrix starts, which initiates drug release.”
- the drug spectrum must be reported in the diffractogram
Answer:
The XRD diffractogram of the synthesized sulfanilamide was added.
- since the drug was synthesized de novo, a more in-depth characterization must be reported e.g. 1H-NMR
Answer:
The complete characterization of the synthesized sulfanilamide by applying FTIR, UV-Vis, XRD, DSC, SEM, 1H-NMR and UHPLC-DAD-ESI-MS/MS methods was performed. We added the 1H NMR spectrum of synthesized sulfanilamide into the manuscript.
1H NMR spectrum of synthesized sulfanilamide in D2O
The mass spectrum of synthesized sulfanilamide
- line 143: what do the authors mean by swollen quickly?
Answer:
The sentence was rewritten in the following manner:
“Firstly, the swollen hydrogels were frozen at -40 °C for 24 h.”
- the authors demonstrated that the purification procedure is effective in eliminating uncrosslinked monomers. however, no analysis has been conducted to demonstrate that there are no traces of methanol.
Answer:
Methanol was washing was performed with methanol/water solutions with an increase in the water content up to 100% (lines 257-260) and then drying of the hydrogel to xerogel. No methanol analyzes were performed, but it is assumed that no methanol can remain in the gel after this washing and drying procedure.
- the procedure for drug incorporation needs to be described in more detail. have they been washed? if yes, how many and with which solvent. How was the methanol eliminated?
Answer:
The requested details were added. The section Sulfanilamide incorporation into the hydrogels was rewritten in the following manner:
“In order to incorporate sulfanilamide into the polymeric network, synthesized xerogels (~0.05 g) were swelled in the sulfanilamide methanolic solution concentration 50 mg/cm3 for 48 h at room temperature. The available amount of sulfanilamide for incorporation into the hydrogel was 500 mg/gxerogel. After reaching equilibrium, the swollen p(NIPAM) hydrogels with incorporated sulfanilamide were separated from the solution by decanting. The hydrogel samples were washed using distilled water to remove excess sulfanilamide. The masses of samples before and after swelling in the sulfanilamide solution were measured in order to calculate the loading efficiency. Loading efficiency (η) of sulfanilamide was calculated using Equation (5):
? (%)= ??/?? ∙100 (5)
where Lg is the mass of sulfanilamide incorporated into the hydrogel (mg/gxerogel) and Lu is the maximum sulfanilamide mass available for incorporation (mg/gxerogel).”
- Lines 210-211: authors should specify the medium.
Answer:
The sentence is rewritten in the following manner:
“The swollen hydrogels with incorporated sulfanilamide were soaked in 3 cm3 of adequate medium (a solution of hydrochloric acid pH 2.2 or sodium hydroxide pH 7.4) and tempered in a water bath at 37±1 ºC for 24 h.”
- Swelling results lack of statistical analysis. Also, the reason why 1.5% hydrogel shows higher swelling is poorly explained.
Answer:
Error bars were added to the figures. The authors are grateful for the suggestion.
The hydrogel synthesized with 1% of crosslinker showed that the created polymer network is not optimal for reaching a high swelling degree. The assumption is that some parts of the polymer network have too long branches and too few network nodes due to the low concentration of crosslinker. These parts of the polymer network partially dissolve and do not contribute to the swelling degree. Hydrogel with 1.5% crosslinker has a higher degree of swelling due to a favorable ratio of branch lengths and network nodes. With the further increase in the concentration of crosslinker, the branches are shortened and the concentration of network nodes increases, which reduces the possibility of expansion of the network structure, thus decreasing the swelling degree.
- Since the main stimulus that influences the hydrogel swelling behavior is the temperature, release experiment must be conducted at different temperatures (25 and 37°C). This is crucial to demonstrate that entire rationale of the work.
Answer:
The homopolymer p(NIPAM) has a lower critical solution temperature (LCST) of 32 °C (Heskins and Guillet, 1968). The manuscript shows the release of sulfanilamide at 37 °C (which is above the LCST). Also, according to the suggestion of the reviewer, the release results at 20 °C (which is below the LCST) were added.
Submission Date
15 April 2023
Date of this review
05 May 2023 13:04:18

Reviewer 5 Report
The author prepared a drug delivery matrix based on the use of PNIPAAM hydrogel. Overall, the concept is general and quite similar with any reported journal but the measurement and the approach is quite different. We though that this manuscript could be improved and several adjustment has to be conducted before publication.
1. Why did the author measure the drug release at pH 2.2? If we use the extracellular or intracellular condition of the cells, I suggest the author to measure the release profile at the range of 5-7.4.
2. Why did the system (1%) show burst release from 4 h to 24 h?
3. A comparison with reported result should be added.
4. Some of the images are blurred, please improve the quality.
5. The author describes about the change in the volume. Did the author investigate the change in the hydrogel’s volume after or during swelling?
6. If by any chance, this system will be used for drug delivey, a biocompatibility test has to be performed.
Several mistyping and missing punctuation can be found in the manuscript. We assume that an english revision should be performed.
Author Response
Comments and Suggestions for Authors
The author prepared a drug delivery matrix based on the use of PNIPAAM hydrogel. Overall, the concept is general and quite similar with any reported journal but the measurement and the approach is quite different. We though that this manuscript could be improved and several adjustment has to be conducted before publication.
- Why did the author measure the drug release at pH 2.2? If we use the extracellular or intracellular condition of the cells, I suggest the author to measure the release profile at the range of 5-7.4.
Answer:
The release of sulfanilamide from p(NIPAM) hydrogels of different crosslinking degree was monitored in the solutions of pH of 2.2 and 7.4 at temperature of 37±1 °C in order to simulate conditions in the different parts of the gastrointestinal tract (gastric and intestinal pH environment). The similar analysis conditions were used in the available literature (https://doi.org/10.3390/pharmaceutics13020158; https://doi.org/10.3390/polym1403052).
- Why did the system (1%) show burst release from 4 h to 24 h?
Answer:
The hydrogel with 1% crosslinker showed some irregularity in the degree of swelling compared to the other samples. The authors interpret this phenomenon as the incomplete formation of the network, so that the hydrogel with 1.5% crosslinker shows the highest degree of swelling. Probably, this anomaly is also reflected in the release of sulfanilamide from the incompletely formed gel.
- A comparison with reported result should be added.
Answer:
We added the paragraph with a comparison with the reported results.
- Some of the images are blurred, please improve the quality.
Answer:
We improved the quality of the figures. All of the figures are in resolution of 1200 dpi.
- The author describes about the change in the volume. Did the author investigate the change in the hydrogel’s volume after or during swelling?
Answer:
The authors investigated the change of hydrogel volume during swelling.
- If by any chance, this system will be used for drug delivey, a biocompatibility test has to be performed.
Answer:
This manuscript represents one part of the planned and performed investigations of synthesized p(NIPAM) hydrogels as potential carriers for the antimicrobial active substance sulfanilamide.
Comments on the Quality of English Language
Several mistyping and missing punctuation can be found in the manuscript. We assume that an english revision should be performed.
Answer:
The manuscript was revised by a native English speaker.
Submission Date
15 April 2023
Date of this review
02 May 2023 04:13:25

Round 2
Reviewer 1 Report
If the PNIPAM hydrogel cannot be used as smart drug delivery system (because the released of sulfanilamide is not strongly influenced by the temperature) what is the advantage of using such hydrogels for the delivery of sulfanilamide? The novelty of the work is very low.
If the justification of the paper was that “In direct and indirect contact with mouse fibroblasts, p(NIPAM) hydrogel does not reduce the viability of fibroblast cells and thus shows minimal cytotoxicity”, in-vitro and in vivo tests should be perform to test the biocompatibility of the present hydrogels loaded with sulfanilamide.
Some minor apects can be re-considered:
If it is possible, the SEM images without and with loaded drug should have the same scale.
Line 488: If no peaks from sulfanilamide crystals were observed in the XRD spectrum of loaded hydrogel, how „Irregularly distributed sulfanilamide crystals were observed” in the SEM imges?
Line 553: It is written „as well as the partial collapse of the hydrogel at a lower temperature”, but the hydrogel collapsed at higher temperatures, as shown in the Figure 5.
Lines 556-557: Ref 47 does not contain drug release studies. Ref 48 contains release studies of piroxicam only at 37 °C. Ref 49 contains release studies of naproxen only at 38 °C, and ref 50 deals with release of phencetin at 37 °C. Only reference 51 presents the release of ibuprofen at 20 C and 40 C, and at 40 C the release of the drug is higher due to the collapse of the matrix. In the case of the present work, the release of the drug is the same below and above the VPTT, even if the swelling at this temperatures ( 20 and 37 C) are very different. It would be helpful to introduce references showing that the release in PNIPAM hydrogels is not influence by the temperature.
-
Author Response
Comments and Suggestions for Authors
If the PNIPAM hydrogel cannot be used as smart drug delivery system (because the released of sulfanilamide is not strongly influenced by the temperature) what is the advantage of using such hydrogels for the delivery of sulfanilamide? The novelty of the work is very low.
The results of the release of sulfanilamide from this hydrogel showed about 25-30% higher drug delivery at a temperature of 37±1°C compared to 20±1°C. The advantage of using this hydrogel is that the entire amount of sulfanilamide is not released, but is released slowly over a longer period of time.
If the justification of the paper was that “In direct and indirect contact with mouse fibroblasts, p(NIPAM) hydrogel does not reduce the viability of fibroblast cells and thus shows minimal cytotoxicity”, in-vitro and in vivo tests should be perform to test the biocompatibility of the present hydrogels loaded with sulfanilamide.
The authors agree with this statement that the cytotoxicity of the formulation should be done in vitro and in vivo, regardless of the fact that the biocompatibility of the gel was studied in publication 17 (doi.org/10.1002/jbm.a.32322) and sulfanilamide has been in human use for about 85 years. Any new formulation must be subjected to these tests. However, the authors are not in a position to do these tests in a short time, especially in vivo, which require the permission of the ethics committee, a process that takes several months. With this manuscript, the authors wanted to perform chemical tests on the possibility of making such a formulation. Cytotoxicity may be the subject of extensive further investigation.
Some minor apects can be re-considered:
If it is possible, the SEM images without and with loaded drug should have the same scale.
Unfortunately, it is not possible. The authors do not have other SEM images because the research was done earlier. On the other hand, the authors believe that the difference in scale is not so great that the necessary details cannot be seen on them.
Line 488: If no peaks from sulfanilamide crystals were observed in the XRD spectrum of loaded hydrogel, how „Irregularly distributed sulfanilamide crystals were observed” in the SEM imges?
Table 3 shows that the content of sulfanilamide in the xerogel is less than one third of the total mass. It should be taken into account that in the process of drying the gel with sulfanilamide, sulfanilamide began to separate from the swollen gel in the form of clusters of molecules. Mixed with a gel that also dries and changes volume, it could not form pure and large crystals. Thus, on SEM photographs, adhered clusters are observed as a multitude of small crystals far from each other, which is why any pure X-ray reflection that would be detected is impossible.
The solubility of sulfanilamide in water is 7.5 mg/ml and in methanol about 27 mg/ml. Thus, a larger amount of sulfanilamide was introduced with methanol than can later be dissolved in the water in which the hydrogel swells. Therefore, the amount of sulfanilamide in the hydrogel is above the solubility limit, and molecular clusters are deposited in the pores of the gel.
Line 553: It is written „as well as the partial collapse of the hydrogel at a lower temperature”, but the hydrogel collapsed at higher temperatures, as shown in the Figure 5.
The authors apologize, the construction of the sentence is not good. The authors wanted to say that there was less collapse of the gel at a lower temperature compared to the intensity of the collapse that occurs to the gel at a higher temperature. Thus, even that smaller collapse contributed somewhat to the release of sulfanilamide, but less than at a higher temperature. The manuscript has been corrected.
Lines 556-557: Ref 47 does not contain drug release studies. Ref 48 contains release studies of piroxicam only at 37 °C. Ref 49 contains release studies of naproxen only at 38 °C, and ref 50 deals with release of phencetin at 37 °C. Only reference 51 presents the release of ibuprofen at 20 C and 40 C, and at 40 C the release of the drug is higher due to the collapse of the matrix. In the case of the present work, the release of the drug is the same below and above the VPTT, even if the swelling at this temperatures ( 20 and 37 C) are very different. It would be helpful to introduce references showing that the release in PNIPAM hydrogels is not influence by the temperature.
When making corrections in the manuscript with the Track Changes option enabled, the authors did not notice that there was an error in the order of cited references. Thus, the references were ranked one higher. The authors made a mistake by citing the papers that resulted from the PhD of Snežana Ilić Stojanović, in which all these investigations are found at temperatures above and below the LCST, but all the results were not included in the publications. Therefore, these references will be removed except for the ibuprofen reference. In that publication, there is a similar tendency to release the active substance from the hydrogel. Namely, at a lower temperature, about 30% less ibuprofen is released in the same time. In this publication also about 25-30% less amount of sulfanilamide was released at lower temperature. This has been corrected in the manuscript. The difference is not big, but the tendency is similar. It is probably about mechanisms that we have not yet fully elucidated.
The authors are grateful for helpful suggestions and advice.
Comments on the Quality of English Language
-
Submission Date
15 April 2023
Date of this review
23 May 2023 16:19:41
Reviewer 2 Report
Unsolved Comment #1: Introduction section: Authors commented that previous works have reported the preparation and evaluation of sulfanilamide-polymer carriers for the controlled release of this drug. The motivation and the novelty of the present research should be further highlighted. It is not enough to say that it is the first time that the p(NIPAM)/ sulfanilamide system is studied.
Please, include some additional comment to highlight the novelty of the research.
Unsolved Comment #2: The research core of this work was the development of PNIPAAm-based hydrogels for controlled drug delivery. To balance the scientific viewpoint and attract more attention from audiences, the authors are highly recommended to consider the inclusion of the recent relevant papers, such as: https://doi.org/10.3390/pharmaceutics13081284
It is strongly recommended the inclusion of the reported manuscript as support of the discussion of present research.
Author Response
Comments and Suggestions for Authors
Unsolved Comment #1: Introduction section: Authors commented that previous works have reported the preparation and evaluation of sulfanilamide-polymer carriers for the controlled release of this drug. The motivation and the novelty of the present research should be further highlighted. It is not enough to say that it is the first time that the p(NIPAM)/ sulfanilamide system is studied.
Please, include some additional comment to highlight the novelty of the research.
The situation is really like that. Copolymer hydrogels have mainly been used for the delivery of some drugs: a hydrogel based on NIPAM and dextran was used as a carrier for the controlled release of ornidazole and ciprofloxacin (Das et al., 2015), while nanoparticles made from hydrogels based on NIPAM, acrylic acid and hydroxyethyl methacrylate formulated to achieve controlled release of amoxicillin (Moogooee et al., 2011). Copolymer hydrogels of p(NIPAM) with gelatin (Manjula et al., 2014), methacrylic acid (Zafar et al., 2014) or allyl-amine (James et al., 2011) can be used as carriers for silver nanoparticles, which are used in antimicrobial therapy of wounds and burns. Copolymer hydrogels of NIPAM with 2-hydroxypropyl methacrylate were investigated as carriers for the controlled release of ibuprofen (Ilić-Stojanović et al., 2013).
Lee and Lee synthesized hybrid hydrogels based on NIPAM and gelatin for encapsulation and release of sulfanilamide from hybrid hydrogels (Lee and Lee, 2007). Lee and Huang investigated the release of sulfanilamide, as a model drug, from a copolymer hydrogel based on NIPAM, glycidyl methacrylate (GMA) and N,N-dimethylacrylamide (DMA) (Lee and Huang, 2008). Lee and Cheng investigated the effect of monomer content on sulfanilamide release and properties of a biodegradable hydrogel based on NIPAM, acrylic acid, and poly(caprolactone)-diacrylate (Lee and Cheng, 2013).
These researches were the motive to carry out tests of the homopolymer p(NIPAM) with sulfanilamide and define the possibility of making such formulations as well as their potential application.
This analysis is included in the manuscript. The authors are grateful for the suggestion.
Unsolved Comment #2: The research core of this work was the development of PNIPAAm-based hydrogels for controlled drug delivery. To balance the scientific viewpoint and attract more attention from audiences, the authors are highly recommended to consider the inclusion of the recent relevant papers, such as: https://doi.org/10.3390/pharmaceutics13081284
It is strongly recommended the inclusion of the reported manuscript as support of the discussion of present research.
The authors are grateful for the suggestion. A discussion of the proposed reference has been added to the manuscript.
Submission Date
15 April 2023
Date of this review
22 May 2023 19:45:51
Reviewer 4 Report
Authors did not respond satisfactorily to some requests. In particular, they don't address the pH issue. So here's the original comment:
1. since the starting polymer has no ionizable functional groups it is clear that the swelling cannot depend on the pH. Please explain the reason of conducting characterization at different pH.
2. The authors cannot simply assume that there is no residual methanol. It should be demonstrated
3. Authors state that “the intermolecular interactions between sulfanilamide and side groups of the polymer matrix get broken and contraction of the polymer matrix starts, which initiates drug release”
However since SEM analyses clearly indicates the presence of crystals inside the hydrogel, so this explanation is not corroborated by results. The drug should be in molecular state inside the hydrogel to establish intermolecular interactions.
Author Response
Comments and Suggestions for Authors
Authors did not respond satisfactorily to some requests. In particular, they don't address the pH issue. So here's the original comment:
- since the starting polymer has no ionizable functional groups it is clear that the swelling cannot depend on the pH. Please explain the reason of conducting characterization at different pH.
The authors thank you for the suggestion and agree that without ionizing groups, the pH change has no significant effect on the properties of the hydrogel. However, research was conducted at pH 2.2 due to potential oral administration and similar acidity in the stomach. pH 7.4 was chosen as the pH of other physiological fluids. If the reviewer insists that the results at pH 2.2 not be shown, the authors will do so but they consider that the work with the results at pH 2.2 is more complete.
- The authors cannot simply assume that there is no residual methanol. It should be demonstrated
The authors extracted the dried gel (xerogel) for 24 h with distilled water. The sample was filtered on a 0.45 µm PTFE filter and analyzed by HPLC Agilent 1100 Series (Waldborn, D) equipped with refractive index detector, RID 1200 Series. Conditions for chromatography performance: column Zorbax Eclipse XDB-C18 (4.6 × 250 mm, 5 μm) (Agilent Technologies, Inc., Santa Clara, CA, USA); temperature 25 °C; injected volume of samples 20 μL; mobile phase was redistilled water; mobile phase flow was 1 cm3/min. The results obtained were processed by software Agilent Chemstation.
The graphic shows a chromatogram that is at the noise level and has no peak from methanol (very small nRIU values are visible). Although the authors checked this, they considered that such graphics in the manuscript are not appropriate and that it is enough to state that there is no methanol in the gel. This is understandable after drying the gel, from which even traces of methanol evaporate easily.
When there are some substances in the sample, peaks with large nRIU values appear because the RID detector is sensitive enough. An example is the following graph showing nRIU values over 25000:
- Authors state that “the intermolecular interactions between sulfanilamide and side groups of the polymer matrix get broken and contraction of the polymer matrix starts, which initiates drug release”
However since SEM analyses clearly indicates the presence of crystals inside the hydrogel, so this explanation is not corroborated by results. The drug should be in molecular state inside the hydrogel to establish intermolecular interactions.
The solubility of sulfanilamide in water is 7.5 mg/ml and in methanol about 27 mg/ml. Thus, a larger amount of sulfanilamide was introduced with methanol than can later be dissolved in the water in which the hydrogel swells. Therefore, the amount of sulfanilamide in the hydrogel is above the solubility limit, and molecular clusters are deposited in the pores of the gel.
Intermolecular interactions can also be established when the drug is in the form of a cluster of molecules. Since these are small groups of sulfanilamide molecules in the gel, the molecules on the surface of those microcrystals certainly interact with the surface of the gel with which they are in contact. This will later be reflected in the release profile of sulfanilamide from the gel.
Drug release from hydrogels can be diffusion controlled, swelling controlled or chemically controlled (Lin and Metters, 2006). The rate of release and the amount of drug released from the hydrogel depend on the degree of swelling and crosslinking density of the hydrogel, as well as on the properties of the drug itself, such as molecular weight and charge (Lee and Lee, 2007). In p(NIPAM)-based hydrogels, the initial rapid release of the active substance is followed by a much slower, controlled release (Ashraf et al., 2016). The authors thank you for the suggestion.
Submission Date
15 April 2023
Date of this review
22 May 2023 17:28:56

Reviewer 5 Report
Thank you for answering the question. The arrangement of the figure is poor such as the NMR, which is too big for 1 page.
Author Response
Comments and Suggestions for Authors
Thank you for answering the question. The arrangement of the figure is poor such as the NMR, which is too big for 1 page.
The authors are grateful for the suggestions. Unfortunately, the authors have no possibility to further improve the quality of the 1H-NMR image of sulfanilamide.
Submission Date
15 April 2023
Date of this review
22 May 2023 07:24:11
Round 3
Reviewer 1 Report
The paper was improved somehow.
If the release of sulfanilamide is slower at 20 °C compared to 37 °C, and "the advantage of using this hydrogel is that the entire amount of sulfanilamide is not released, but is released slowly over a longer period of time", it means that you recommend this hydrogels for drug release at 20 °C?
The results of the release of sulfanilamide from this hydrogel showed about 25-30% higher drug delivery at a temperature of 37±1°C compared to 20±1°C. .
See the introduced sentence from lines 130-132.
See the modified sentence from lines 556-558.
Author Response
Comments and Suggestions for Authors
The paper was improved somehow.
If the release of sulfanilamide is slower at 20 °C compared to 37 °C, and "the advantage of using this hydrogel is that the entire amount of sulfanilamide is not released, but is released slowly over a longer period of time", it means that you recommend this hydrogels for drug release at 20 °C?
The results of the release of sulfanilamide from this hydrogel showed about 25-30% higher drug delivery at a temperature of 37±1°C compared to 20±1°C. .
The authors are grateful for this suggestion. Although much work and analysis remains before any use, the authors think that 37°C is a relevant temperature for drug release for human use and that further research should focus on 37°C.
Comments on the Quality of English Language
See the introduced sentence from lines 130-132.
See the modified sentence from lines 556-558.
The authors wanted to say that the collapse of the hydrogel allows a smaller contribution to the release of sulfanilamide at a lower temperature than the contribution of the collapse of the hydrogel at a higher temperature. But, at the same time, at a higher temperature, the contribution of diffusion is greater and thus the overall effect is a greater release of sulfanilamide at a higher temperature. The authors are grateful for the suggestion and they corrected the manuscript.
Submission Date
15 April 2023
Date of this review
29 May 2023 19:12:24
Reviewer 4 Report
the authors responded satisfactorily to the requests.
Author Response
Comments and Suggestions for Authors
The authors responded satisfactorily to the requests.
The authors are grateful for their suggestions and comments.
Submission Date
15 April 2023
Date of this review
29 May 2023 09:07:08